*Report*

EMBO
Molecular Medicine

# Proteomics and personalized PDX models identify treatment for a progressive malignancy within an actionable timeframe

Georgina D Barnabas[1,2,17], Tariq A Bhat [2,3,17], Verena Goebeler [2,3], Pascal Leclair[2,3], Nadine Azzam[4], Nicole Melong[4], Colleen Anderson[5,6], Alexis Gom[5,7], Seohee An[2,3], Enes K Ergin[1,2], Yaoqing Shen[8], Agustina Conrrero[1], Andrew J Mungall [8], Karen L Mungall[8], Christopher A Maxwell [2,3], Gregor S D Reid [2,3], Martin Hirst[8,9], Steven Jones [8,9], Jennifer A Chan [5,6,10], Donna L Senger [11,12], Jason N Berman [4,13], Seth J Parker [14,15], Jonathan W Bush [1], Caron Strahlendorf[3], Rebecca J Deyell [2,3✉], C James Lim [2,3✉] & Philipp F Lange [1,2,16✉]

## Abstract

Genomics has transformed the diagnostic landscape of pediatric malignancies by identifying and integrating actionable features that refine diagnosis, classification, and treatment. Yet, translating precision oncology data into effective therapies for hard-to-cure childhood, adolescent, and young adult malignancies remains a significant challenge. We present the case for combining proteomics with patient-derived xenograft models to identify personalized treatment for an adolescent with primary and metastatic spindle epithelial tumor with thymus-like elements (SETTLE). Within two weeks of biopsy, proteomics identified elevated SHMT2 as a target for therapy with the anti-depressant sertraline. Drug response was confirmed within two months using a personalized chicken chorioallantoic membrane model of the patient's SETTLE tumor. Following failure of cytotoxic chemotherapy and second-line therapy, the patient received sertraline treatment and showed decreased tumor growth rates, albeit with clinically progressive disease. We demonstrate that proteomics and fast-track xenograft models provide supportive pre-clinical data in a clinically meaningful timeframe to impact clinical practice. By this, we show that proteome-guided and functional precision oncology are feasible and valuable complements to the current genome-driven precision oncology practices.

**Keywords** Pediatric Cancer; Proteomics; Genomics; Patient Derived Xenografts; Precision Therapeutics

**Subject Categories** Cancer; Proteomics

## Introduction

Despite diagnostic and therapeutic advancements, cancer remains a leading cause of mortality in children, adolescents, and young adults (CAYA), with an estimated 400,000 new cases and 100,000 deaths each year worldwide (Committee, 2023; McLoone et al, 2023; Schulpen et al, 2021). The molecular landscape of CAYA cancers differs significantly from adult, underscoring the need for CAYA-specific molecular diagnostics (Lorentzian et al, 2018) and posing challenges in translating precision oncology data into effective therapies (Downing et al, 2012; McLeod et al, 2021). Genome-driven precision oncology maps causal alterations and target selection is often facilitated by the identification of molecular similarity to patient cohorts with well-established targeted therapies. However, most drugs target proteins, so proteomics provides a quantitative assessment of cellular processes that provides complementary insight closer to the disease and therapy response phenotype. Though still at an early stage, integrating global proteomic analysis into precision oncology holds promise, particularly for personalized target identification (Nishizuka and Mills, 2016; Wahjudi et al, 2021).

[1]Department of Pathology, University of British Columbia, Vancouver, BC, Canada. [2]Michael Cuccione Childhood Cancer Research Program, BC Children's Hospital Research Institute, Vancouver, BC, Canada. [3]Department of Pediatrics, University of British Columbia, Vancouver, BC, Canada. [4]Children's Hospital of Eastern Ontario (CHEO) Research Institute, Ottawa, ON, Canada. [5]Arnie Charbonneau Cancer Institute, Cumming School of Medicine, University of Calgary, Calgary, AB, Canada. [6]Department of Pathology, Cumming School of Medicine, University of Calgary, Calgary, AB, Canada. [7]Department of Oncology, Cumming School of Medicine, University of Calgary, Calgary, AB, Canada. [8]British Columbia Cancer Agency, Canada's Michael Smith Genome Sciences Centre, Vancouver, BC, Canada. [9]Department of Medical Genetics, University of British Columbia, Vancouver, BC, Canada. [10]Alberta Children's Hospital Research Institute, University of Calgary, Calgary, AB, Canada. [11]Department of Medicine, McGill, Montreal, QC, Canada. [12]Lady Davis Institute for Medical Research, Jewish General Hospital, Montreal, QC, Canada. [13]Department of Cellular and Molecular Medicine, University of Ottawa, Ottawa, ON, Canada. [14]Department of Biochemistry, University of British Columbia, Vancouver, BC, Canada. [15]Centre for Molecular Medicine and Therapeutics, University of British Columbia, Vancouver, BC, Canada. [16]Department of Molecular Oncology, British Columbia Cancer Research Centre, Vancouver, BC, Canada. [17]These authors contributed equally: Georgina D Barnabas, Tariq A Bhat.✉E-mail: RDeyell@cw.bc.ca; c.james.lim@ubc.ca; philipp.lange@ubc.ca

Genome-driven precision oncology trials focussing on CAYA patients have been initiated after their adult counterparts and only recently started to report initial findings. For example, in the MAPPYACTS trial, 56% of children had actionable alterations, but only 16% of those receiving targeted therapy (19%) had partial or complete response (Berlanga et al, 2022). The INFORM registry reported 48% of cases with actionable alterations with 16% receiving targeted therapy (van Tilburg et al, 2021). In a recent INFORM study, patients receiving matched targeted therapies with ALK, NTRK, and BRAF inhibitors experience a significant survival benefit compared to those receiving conventional or no treatment (Heipertz et al, 2023). The ZERO PRISM trial reported a 36% objective response rate and improved progression-free survival for patients receiving precision guided therapy compared to standard of care (Lau et al, 2024). The pediatric Personalized OncoGenomics program achieved a clinical benefit rate of 54% for 32 molecularly informed therapies administered to 28 participants (Deyell et al, 2024). Of 1000 patient enrollments, the ongoing NCI COG Pediatric MATCH trial found actionable mutations in 31% of tumors sequenced, with 28% assigned to treatment arms (Parsons et al, 2022). For reference, the adult-based NCI-MATCH trial achieved an overall response rate of 10.3% out of 6000 patients enrolled (O'Dwyer et al, 2023).

PRecision Oncology For Young peopLE (PROFYLE) is a collaborative pan-Canadian project aiming to provide more effective treatment options to CAYA with advanced cancers. In parallel to genome-driven precision oncology, the prospects of proteome-guided and functional precision oncology approaches are being explored in PROFYLE.

To evaluate if proteome-guided precision oncology can provide de-novo target prioritization in the absence of molecular grouping, here we present the case of a child with a rare and challenging CAYA malignancy, a spindle epithelial tumor with thymus-like elements (SETTLE) with progressive disease. SETTLE originates from ectopic thymus tissue or branchial pouch remnants in the thyroid, and presents a distinct biphasic pattern of spindle and epithelial cells. Although SETTLE is categorized as a slow growing malignancy with a 5-year overall survival of >80%, metastasis has been reported in up to 41% of patients with at least 5-year follow-up, with a latency period of up to 10 years (Grushka et al, 2009; Recondo et al, 2015). Molecular pathogenesis and clinical outcomes of SETTLE remain poorly understood due to limited case studies (Stevens et al, 2019), which provide valuable clinical and histological insights but lack comprehensive molecular characterization.

For the case of a child presenting with three metastatic recurrence of a SETTLE tumor, we hypothesized that quantitative whole proteome profiling and drug response validation could provide timely pre-clinical insight complementary to whole exome and transcriptome sequencing and inform clinical decision-making. To overcome challenges arising from the limited quantity of available fresh or viably cryopreserved tissues that limit the utility of the traditional immune-incompetent mouse models (Abel et al, 2021; Hidalgo et al, 2014), we incorporated the embryonic chick chorioallantoic membrane (CAM) and larval zebrafish models into the study as robust and readily accessible in vivo systems distinguished by their rapid experimental turnover (Keller and Murtha, 2004; Kunz et al, 2019; Pawlikowska et al, 2020; Ribatti,

2016; Tang et al, 2014). By integrating proteomics with PDX modeling, our study reveals potential personalized treatment options that were not evident from genome and transcriptome analyses. This finding shows the value of a combined pre-clinical approach to inform medical decisions for the individual patient, and also encourages accelerated integration of proteome and functional approaches into precision oncology pipelines.

# Results and discussion

## Patient history

At initial diagnosis the child had presented with asymptomatic right neck swelling, which had been incidentally noted. Ultrasound showed a $3.8 \times 2.3 \times 4.5$ cm heterogeneous soft tissue mass replacing the right thyroid lobe, with no associated adenopathy. The patient underwent right hemithyroidectomy confirming a SETTLE tumor (Matheson et al, 2019). This diagnosis started ongoing cycles of testing, and treatment, including genome profiling-guided targeted therapy, period of remission, and recurrence. The patient was enrolled in the PROFYLE study when they experienced a first recurrence or relapse (R1) with metastatic progressive disease (Fig. 1A).

At R1, we conducted whole genome and transcriptome profiling and the patient received sorafenib (Appendix Supplementary Information – R1 Genome Profiling). At R3, we conducted prospective real-time LC-MS/MS-based quantitative whole proteome profiling, candidate target identification, and IHC target validation within 2 weeks of biopsy. To expand the molecular etiology, we also conducted retrospective proteome profiling of biopsies from the primary tumor and recurrences R1 and R2. We used viably cryopreserved R2 tissue to establish and evaluate drug response as CAM PDXs. In parallel, R2 tumor was also established as mouse PDXs, affording expansion and archiving of SETTLE cells for additional CAM and zebrafish ex vivo drug studies, and to confirm on-target activity by isotope tracing in vitro. Whole exome/transcriptome sequencing was performed at R3 to identify actionable alterations not present at R1. Two months after biopsy, the combined pre-clinical insights were evaluated by the PROFYLE Molecular Tumor Board (MTB) followed by initiation of innovative therapy (Fig. 1A,B).

## Multi-omics molecular profiling at third recurrence (R3)

Following the lack of response to conventional chemotherapy, and acquired resistance to sorafenib, the criteria for progressive disease was attained. Additional pulmonary resection was performed (Fig. 1C) and followed by multi-omics analyses, for the first-time including proteome analyses, to identify additional molecular targets and therapeutic options (Fig. 1B). Whole exome sequencing and RNAseq analysis of fresh frozen samples from new thoracoscopic biopsies of pulmonary nodules (R3) revealed tumor genome evolution since R1. In particular, R3 showed increased mutational burden (0.77 mut/Mb) and increased presentation of mutation signature after treatment compared to R1 (0.37 mut/Mb). R3 carried the same notable variants previously identified in R1. These included FLT3 p.L780F, a copy loss and truncating mutation in ARID1A:p.R1335*, a subclonal mutation in SF3B1:p.K700E, and a copy loss in MUS81. Gene expression levels were overall similar between R1 and R3 with some exceptions

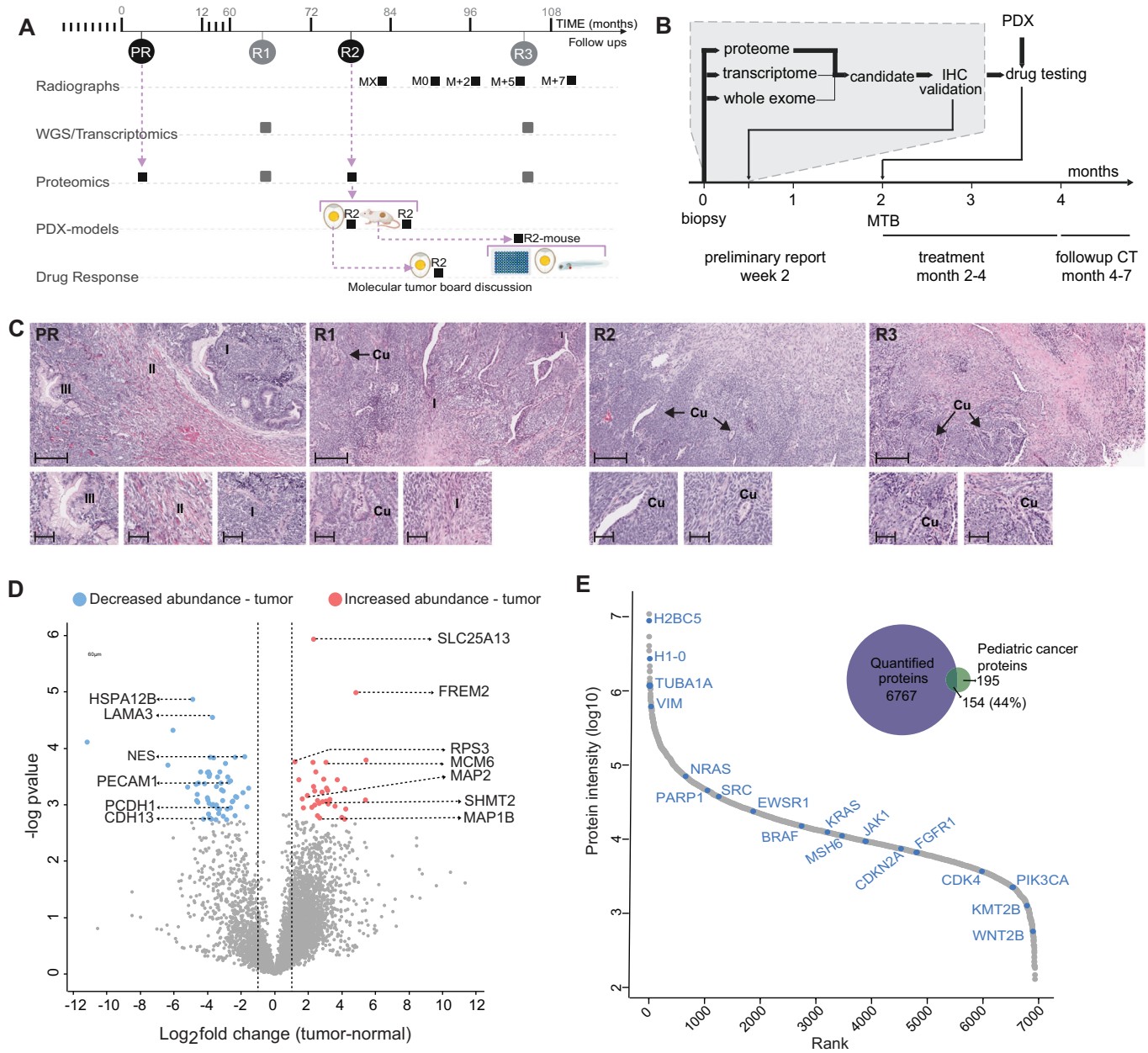

**Figure 1.   Multi-omics molecular profiling of a progressive SETTLE case.**

(A) Patient journey from the resection of the tumor (PR) through three distant recurrences (R1, R2, R3). The molecular analyses, radiographs (Months at time X, MX to M + 7), PDX models, and drug sensitivity assays conducted are linked to the corresponding time points. Proteome analysis of PR, R1 and R2 was performed retrospectively at time of R3. (B) Timeline of precision diagnostics for the SETTLE disease course, including multi-omics molecular tumor profiling, real-time target identification, and validation using personalized xenograft models in providing timely pre-clinical support for medical decision-making. (C) H&E stained sections of the primary resection (PR) and recurrent biopsies (R1-R3). PR demonstrates a biphasic mesenchymal pattern with heterologous elements including gastric foveolar-lined glands showing three patterns, a small blue cell population with a fascicular growth pattern (I). A more collagenous and sclerotic pattern (II). The heterologous epithelial elements show foveolar-like epithelium (III). R1 shows a similar pattern to the resection specimen, including a hypocellular area with increased collagen which transitions into a hypercellular spindle cell area. The cellular nests and fascicles with more collagenous areas (I). Areas of cuboidal-lined (Cu) epithelium are seen. R2 and R3 show hypocellular collagenous tumor transitioning to hypercellular spindled areas with cuboidal-lined (Cu) epithelium. Scale bars: 200 μm in main image; 60 μm in insets. (D) Volcano plot showing the global analysis of proteome perturbation between R3 and adjacent normal regions from three technical replicates for each group. Proteins significant with student's t-test at 0.01 FDR and with log2 fold change >1 are highlighted in red for increased abundance in tumor and in blue for reduced abundance in tumor. (E) Quantified proteins ranked by their average intensity across all samples to evaluate the dynamic range and depth of the proteome coverage. Venn diagram showing the overlap of all proteins quantified in this study with known pediatric cancer-associated proteins. Source data are available online for this figure.

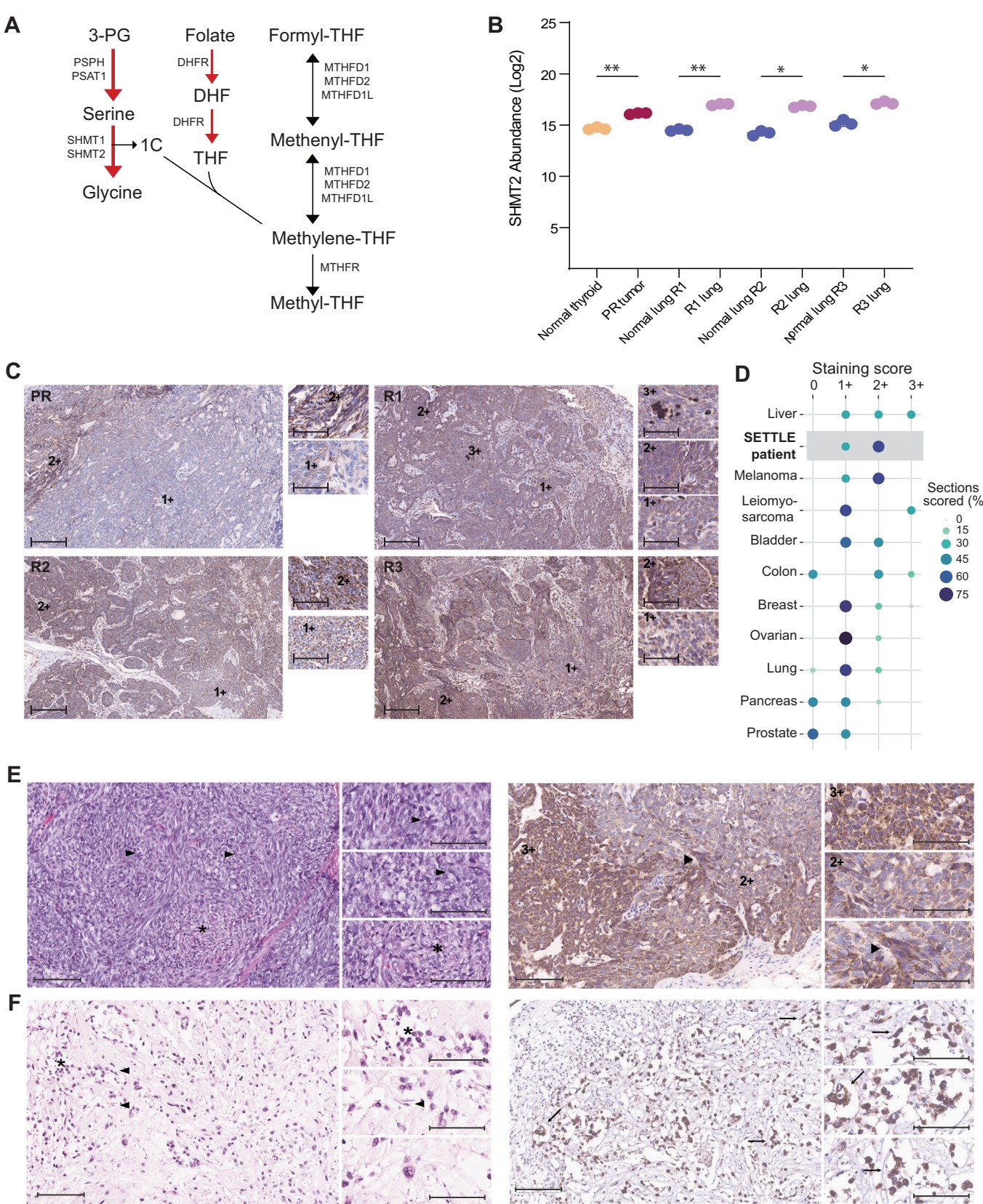

◀  **Figure 2.   Increased one-carbon metabolism and serine addiction in SETTLE tumor.**

(A) Schematic representation of serine biosynthesis and 1C metabolism pathways. Red arrows highlight the increased abundance of proteins involved in serine biosynthesis and those involved in serine and folate metabolism in the SETTLE tumor. (B) Increased abundance of SHMT2 in the primary tumor and lung metastases, data shown represent three technical replicates for each group. Bonferroni's multiple comparisons test, normal thyroid vs PR tumor **p(adjusted) = 0.0069, normal lung R1 vs R1 lung **p(adjusted) = 0.0034, normal lung R2 vs R2 lung *p(adjusted) = 0.0181 and normal lung R3 vs R3 lung *p(adjusted) = 0.0103. (C) Validation of increased SHMT2 protein levels in SETTLE tumor using IHC staining. Adjacent normal thyroid and lung tissues show minimal basal level staining (score +1), tumor cells show more prominent staining (score +2), and granulocytes show the highest staining (score +3). Scale bars: 200 μm in main image; 60 μm in insets. (D) Levels of SHMT2 in diverse tumor types using tumor microarrays; liver ($n = 3$), melanoma ($n = 3$), and leiomyosarcoms ($n = 3$) showed high SHMT2 levels; bladder ($n = 7$), colon ($n = 5$), breast ($n = 14$), ovarian ($n = 12$) and lung ($n = 16$) tumors showed moderate SHMT2 levels and pancreas ($n = 9$) and prostate ($n = 7$) tumors had the lowest SHMT2 levels. (E) Mouse-PDX. H&E staining (left) shows a consistent pattern of cellular and spindle area as seen in the patient-derived specimen (black arrowheads). SHMT2 IHC (right) shows a heterogenous pattern, with a mix of 3+ and 2+ staining in tumor cells. (F) CAM-PDX. H&E staining (left) shows largely discohesive cells with mild to moderate pleomorphism and hyperchromasia, tumor cells are oval-to-spindled in appearance with irregular nuclear membranes and scant cytoplasm. SHMT2 IHC (right) shows the retention of the staining seen in the primary human R2 and mouse-PDX samples. Scale bars: 100 μm in main image; 60 μm in insets. Source data are available online for this figure.

(Fig. EV1A). This included select tyrosine kinases which may explain the acquired resistance to sorafenib (Appendix Supplementary Information—R3 Genome Profiling; Appendix Fig. S1A; Appendix Table S1A,B). Potential options to target the new findings were discussed at the molecular tumor board, but none were deemed clinically actionable at the time.

## Proteomics reveals potentially actionable targets at R3

In parallel to the sequencing analysis, whole proteome profiling (R3) was seamlessly interfaced with routine pathology workflows. Immediately following the biopsy, the anatomical pathology laboratory at BC Children's Hospital prepared H&E-stained sections from clinical FFPE blocks and isolated the tumor and adjacent normal regions by macrodissection. Following transfer to the research laboratory, we used automated sonication-free (ASAP) processing (Barnabas et al, 2023) and mass spectrometry to assemble quantitative proteome maps (Fig. 1D).

We first focussed on proteins associated with routinely used therapies. None of the 29 proteins linked to the 21 targeted agents considered showed significant alterations that supported their prioritization (Fig. EV1B). The only significant change was increased PTEN protein levels in tumor nodules linked to a tumor suppressive effect (Luongo et al, 2019). Alongside PTEN, the PI3K-AKT-mTOR pathway including PIK3CA, AKT2, AKT3, and MTOR showed high RNA expression (Appendix Table S1A,B) but with no significant protein level changes and thus, was considered not clinically actionable by the MTB.

Considering the lack of protein changes supporting established therapies, we investigated the global proteome perturbation and found downregulation of cell adhesion proteins (CDH13, PCDH1) and upregulation of proteins involved in microtubule assembly (MAP1B, MAP2), cell cycle (MCM6), amino acid transport (SLC25A13) and serine metabolism (e.g., serine hydroxymethyltransferase 2 (SHMT2)) (Fig. 1D). The SETTLE tumor may have undergone one-carbon (1C) metabolic reprogramming, with upregulation of SHMT2 suggestive of serine addiction that presented possible therapeutic options (Ward and Thompson, 2012).

## Increased one-carbon metabolism and serine addiction in patient

To solidify the hypothesis of serine addiction and to gain more insights into the dysregulation of proteins and pathways in the

patient's SETTLE tumor, we expanded the global proteome profiling to include retrospective analysis of the primary tumor and earlier lung metastases (R1 and R2). Because we worked with clinical FFPE tissues, it was straightforward to retrieve archival tissue sections from all earlier time points within days. Across the primary tumor, all lung metastases, and adjacent normal lung and thyroid tissue, we quantified 6921 proteins covering an extensive abundance range of five orders of magnitude and including 154 pediatric cancer-associated proteins (Lorentzian et al, 2023), demonstrating in-depth proteome coverage (Fig. 1E) (Appendix Supplementary Information—R3 Proteome Profiling and Proteogenomics; Appendix Fig. S2A–F) and correlation of 0.43 (R1) and 0.42 (R3) with transcript levels that are consistent with RNASeq and MS data analyses from the same tumor (Fig. EV1C,D), as shown in CPTAC studies (Savage et al, 2024; Zhang et al, 2022).

Differential analysis found 795 upregulated proteins and 728 downregulated proteins in primary tumor and lung metastases relative to adjacent normal regions (Appendix Supplementary Information—Differential Proteome and Pathway Analysis; Appendix Fig. S3A–G). As was already observed for analysis of R3, proteins involved in serine biosynthesis and 1C metabolism (Fig. 2A) (Appendix Supplementary Information—One Carbon Metabolism) were also elevated at earlier disease time points. This included phosphoserine phosphatase (PSPH) and phosphoserine aminotransferase 1 (PSAT1) (Fig. EV2A,B). Elevated PSAT1 and PSPH have been identified as indicators of unfavorable prognosis in various tumors (Chan et al, 2020; De Marchi et al, 2017; Feng et al, 2022; Huang et al, 2022; Liao et al, 2019; Liu et al, 2016). Consistent elevation of SHMT2 in the primary tumor and all three lung relapses (Fig. 2B) provided further support to explore SHMT2 as a therapeutic target.

## Orthogonal target validation

Pre-clinical studies in breast and lung cancer PDX models indicate that SHMT2 constitutes an actionable target with the SHMT inhibitor, sertraline (Beekhof et al, 2019; Bernhardt et al, 2017; Geeraerts et al, 2021; Jiang et al, 2018). To obtain orthogonal validation of increased SHMT2 protein in SETTLE cells, we performed IHC staining for SHMT2. The primary tumor and lung metastases showed positive IHC staining for SHMT2 in the tumor cell-enriched regions, whereas adjacent normal thyroid and lung tissues showed minimal staining (Fig. 2C). We also evaluated

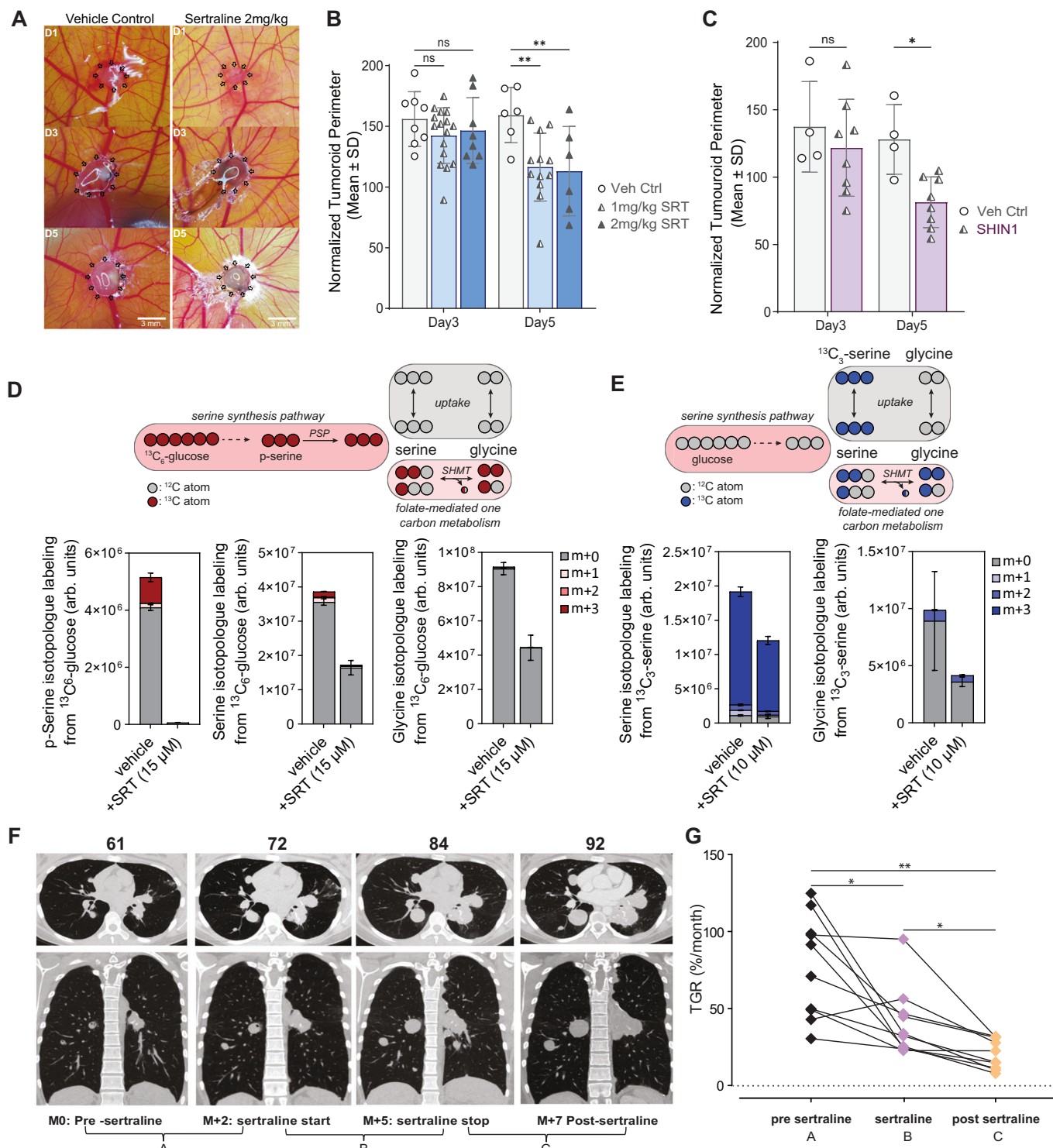

spatial heterogeneity and found that SHMT2 was consistently elevated across the nodules.

Thus far, we had only shown increased abundance of SHMT2 relative to adjacent normal thyroid and lung tissues. In precision oncology studies, target identification is often based on outlier expression relative to larger patient cohorts with prior successful application of the therapy. Sertraline is a common anti-depressant

with good safety profile in pediatric and adult populations (Dwyer and Bloch, 2019; Wagner et al, 2003), and at the time of study, had only recently been identified as an inhibitor of SHMT2 and with pre-clinical PDX data linking SHMT2 levels to drug response (Geeraerts et al, 2021). We therefore used IHC analysis to rank SHMT2 levels in the patient's SETTLE tumor cells with those in tumors previously shown to be susceptible to SHMT2 inhibition in

◀

**Figure 3. Pre-clinical targeting of 1C metabolic pathway in SETTLE and tumor growth rate before and after sertraline therapy.**

(A) Representative images CAM-PDX of SETTLE tumor over five days with sertraline treatment and without (Vehicle Control). Arrowheads indicate periphery of engrafted tumoroids. Scale bars: 3 mm. (B) Measurements of CAM-PDX tumoroid perimeters normalized to day 1 for untreated and treatment with 2 concentrations of sertraline (mean ± s.d., day 3 $n = 8;16;8$ tumors/group, day5 $n = 6;12;6$ tumors/group, two-way ANOVA with Bonferroni's post hoc test, Veh Ctrl vs 1 mg/kg SRT $**p = 0.0041$, Veh Ctrl vs 2 mg/kg SRT $**p = 0.0074$). (C) CAM-PDX tumoroid perimeters, normalized to day 1 for untreated and treatment with SHIN1 (mean ± s.d., day 3 $n = 4;8$ tumors/group, day5 $n = 4;8$ tumors/group, two-way ANOVA with Sidak's post hoc test, Veh Ctrl vs 1 mg/kg SHIN1 $*p = 0.0329$). (D) The abundance of p-serine, serine and glycine isotopologues from $^{13}$C-6-glucose labeled NSG-PDX cells treated with sertraline and the vehicle control (mean ± s.d., $n = 3$ assays per condition). (E) The abundance of serine and glycine isotopologues from $^{13}$C$_3$-serine labeled NSG-PDX cells treated with sertraline or the vehicle control (mean ± s.d., $n = 3$ assays per condition). (F) Patient CT images at 4-time points before, during, and after sertraline therapy spanning 7 months. RECIST score indicated in numbers. (G) Graphs displaying monthly tumor growth rate (TGR) tracked for 9 lesions before (A), during (B) and post (C) sertraline treatment (Data represent %TGR/month, using mixed effects with Bonferroni's post hoc test, Pre-sertraline vs sertraline $*p = 0.0366$, pre-sertraline vs post sertraline $**p = 0.0023$, sertraline vs post sertraline $*p = 0.0146$). Source data are available online for this figure.

diverse pediatric and adult tumor types on tumor microarrays. The patient's tumor manifested high SHMT2 levels similar to liver tumor, melanoma, and leiomyosarcoma, which had the highest number of tissue sections with increased staining (Fig. 2D) (Appendix Fig. S3H). Bladder, colon, breast, ovarian and lung tumors showed moderate SHMT2 staining, and pancreas and prostate tumors had the lowest intensities. Breast tumors had been reported to be susceptible to SHMT2 inhibition (Geeraerts et al, 2021), supporting the finding that the SHMT2 elevation found in this patient might render the malignant cells sensitive to SHMT2 targeted treatment.

## Pre-clinical SHMT2 inhibition in patient derived xenograft models

To evaluate tumor response to SHMT2 inhibition, we established PDX models of R2 SETTLE in CAM and NSG mice (Fig. 1A), and confirmed the presence of histological features by H&E and SHMT2 expression in the PDXs that resembled the clinical sample (Fig. 2E,F) (Appendix Supplementary Information—PDX Models; Appendix Figs. S4, S5). CAM-PDXs treated with SHMT1/2 inhibitors sertraline (Fig. 3A,B) or SHIN1 (Fig. 3C) exhibited reduced tumoroid growth when compared to their respective vehicle controls. To assess the response as a function of serine biosynthesis ability, we evaluated the effects of sertraline treatment on CAM xenografts of serine-addicted MDA-MB-468 and non-addicted MDA-MB-231 tumor cell lines (Geeraerts et al, 2021). As expected, sertraline inhibited MDA-MB-468 tumoroid growth at greater levels than that of MDA-MB-231 (Fig. EV3A,B).

The expanded R2 SETTLE harvested from NSG PDXs afforded material for additional drug testing. In vitro, we determined a half maximal effective sertraline concentrations (IC50) between 6 and 8 µM (Fig. EV3C). Proteomics and pre-clinical studies (Geeraerts et al, 2021) suggested additional druggable opportunities that may be exploited for combination therapy to elicit a more effective response (Appendix Supplementary Information—Drug Combinations). As a monoagent, sertraline or trimethoprim, a DHFR inhibitor (Makino et al, 2022), inhibited the growth of SETTLE cells in CAM-PDX and zebrafish-PDX, but achieved no additional benefit when used in combination (Fig. EV3D,E). While in vitro, SETTLE cells exhibited no response to trimethoprim as a monoagent and negligible benefits in combination with sertraline (Fig. EV3F). In contrast, SETTLE cells exhibited a high degree of synergistic response when sertraline was combined with the mitochondrial inhibitor artemether (Geeraerts et al, 2021), in a

manner resembling the response exhibited by the serine-addicted MDA-MB-468 cells (Appendix Fig. S6).

## Confirmation of serine addiction and validation of on target effect

To validate that the observed response to sertraline was on target and indeed linked to the increased dependence on serine and glycine metabolism in SETTLE cells, we performed stable-isotope tracing using $^{13}$C$_6$-glucose or $^{13}$C$_3$-serine on NSG-PDX cells cultured in vitro. $^{13}$C$_6$-glucose and $^{13}$C$_3$-serine yield m + 3 labeled serine through the serine synthesis pathway activity and uptake, respectively (Fig. 3D,E). Subsequent metabolism by folate-mediated 1C metabolism (FOCM), which includes SHMT1/2, leads to the shuffling of labeled and unlabeled carbons due to the cyclic nature of the pathway. Thus, the relative activity of SHMT1/2 can be inferred by quantifying the abundance of m + 1 and m + 2 versus m + 3 serine isotopologues. Our results from a 12-h culture with $^{13}$C$_6$-glucose suggested that the activity of serine synthesis is relatively low in NSG-PDX cells with sertraline treatment, as m + 3 serine reached only ~5%. In contrast, cells cultured with $^{13}$C$_3$-serine for 24 h displayed ~94% labeling split between m + 1, m + 2, and m + 3 isotopologues and reduced abundance upon sertraline treatment. This suggested that SETTLE cells acquire serine primarily through environmental uptake. Further, the m + 1 and m + 2 isotopologues, which arise through FOCM flux, decreased by ~20% with sertraline treatment (Fig. 3D,E).

## The patient treated with sertraline had a delayed and reduced tumor growth rate

The proteomic analysis and drug response from personalized xenograft models led a molecular tumor board to prioritize a three-month innovative therapy trial with sertraline for the patient. CT scans encompassing the periods before initiation of sertraline therapy, during and after cessation of therapy revealed notable decreases in the monthly tumor growth rates at end of sertraline therapy, which further declined at 2 months post sertraline therapy, with no new parenchymal lesions forming (Fig. 3F). This suggests that the tumor responded to sertraline as a single-agent therapy, slowing the progression of tumor growth, although it did not reduce the tumor mass and RECIST scoring indicating progressive disease (Fig. 3G).

## Conclusion

This case study provides single case evidence that proteomics can provide actionable molecular insights beyond genomic alterations within days to weeks enabling its application in a personalized precision oncology context. Precision medicine programs are enhancing multi-omics in-depth tumor profiling with functional testing of drug sensitivities in patient-derived models to support clinical decisions beyond existing approaches (Irmisch et al, 2021). Rodent-based PDXs have been pivotal in establishing a wide range of models of pediatric solid tumors (Tentler et al, 2012), with the caveat that these xenografts may require months to engraft. While organoid models can provide a rapid alternative for certain tumors (Lampis et al, 2024), the lack of defined protocols critical for the successful generation of patient-derived organoids for CAYA malignancies with a rare occurrence, such as the current SETTLE tumor, prevented this approach. In comparison, CAM and larval zebrafish offer lower-cost and medium-throughput vascularized in vivo platforms for real-time pre-clinical validation of personalized therapies (Kunz et al, 2019; Rebelo de Almeida et al, 2020; Wu et al, 2017) that may be used to bridge and compliment the slow but reliable NSG xenograft models.

We acknowledge this study is limited to a single case, thus it is not able to assess how many and which patient population will benefit the most. To answer these important questions, we are now expanding this to children and adolescents with hard-to-treat cancers across Canada. However, we feel that it is important to report this first case immediately to widely encourage and support similar activities conducted at other pediatric cancer centers. This study sets up the basis to implement proteomics into molecular tumor profiling of patients in precision medicine initiatives. Previous retrospective cohort studies have demonstrated that proteome analysis can identify drug targets and contribute to rational treatment choices in pediatric cancers (Lorentzian et al, 2023; Petralia et al, 2020). Furthermore, the integration of additional data layers, such as single-cell analysis, spatial profiling, and liquid biopsy models, including those considering proteoforms (Barnabas et al, 2023), leverages the multiscale approach to enhance disease understanding and uncover novel biomarkers (Akhoundova and Rubin, 2022). In addition, delving deeper into alternative approaches for combination therapy studies could help identify the optimal fit or achieve better response outcomes, aiding in the prioritization of one combination over another in such scenarios. Therefore, these programs should engage patients at earlier stages and explore combination therapy strategies to improve the number of patients experiencing clinical benefits.

## Significance

This case report emphasizes the benefits of integrating proteomics and personalized xenograft models to verify and complement therapeutic opportunities identified by genomics and expedite the evaluation of treatment responses for rare pediatric malignancies. The ability to provide timely, clinically relevant data invisible to established sequencing-focused precision oncology strategies offers a paradigm shift in medical decision-making and has the potential to significantly influence clinical practice.

## Methods

**Reagents and tools table**

| Reagent/Resource | Reference or Source | Identifier or Catalog Number |
|---|---|---|
| **Experimental models** | | |
| Fertilized chicken eggs (*Gallus gallus domesticus*) | University of Alberta, Edmonton, Alberta, CA | Origin ID: YEGA |
| Casper Zebrafish | University of Ottawa, CA | NA |
| NSG mice | Jackson Laboratories, Bar Harbor, MA, USA | NOD.Cg-Prkdcscid Il2rgtm1Wjl/SzJ |
| MDA-MB-231 | ATCC | HTB-26 |
| MDA-MB-468 | ATCC | HTB-132 |
| **Antibodies** | | |
| SHMT2 (rabbit Polyclone) | Proteintech, USA | 11099-1-AP |
| LAMP1(anti-human) | Biolegend, USA | clone H4A3 |
| biotinylated secondary antibody | Vector Laboratories, USA | VECTASTAIN Elite |
| Isotype IgG control | Vector Laboratories, USA | VECTASTAIN Elite |
| **Reagent/resource** | | |
| DAB+ substrate | (Dako, North America) | NA |
| ShurMount and Cytoseal 60 | Epredia | NA |
| SP3 Magnetic Carboxylate bead | G&E | 451521050250, 651521050250 |
| **Chemical, enzymes and other reagents** | | |
| FBS | Gibco, Waltham, MA | Ref.:12483-020 |
| Geltrex | Gibco, NY, USA | Lot#A1413-02 & A14132-02 |
| RPMI 1640 media | Gibco, Waltham, MA | NA |
| CellTiter-Glo | Promega | NA |
| Trypsin/LysC Mix | Promega | V507A |
| Mouse Cell Depletion Kit | Miltenyi Biotec | NA |
| Dulbecco's phosphate-buffered saline | Gibco, Waltham, MA | NA |
| Dulbecco's phosphate-buffered saline (DPBS) | Sigma-Aldrich | D8537 |
| CoDMEM | Corning | NA |
| penicillin-streptomycin | Invitrogen | NA |
| DMEM | Sigma-Aldrich | RMBN5412 |
| Non-essential amino acids | Invitrogen | NA |
| Sertraline | Selleckchem, USA | S4053 |
| SHIN1 | Tocris | 6998 |
| Dead cell removal kit | Miltenyi Biotec | 130-090-101 |
| DeepRed | ThermoFisher | C34565 |
| DMEM lacking glucose and amino acids | US Biological | D9800-27 |
| HPLC-grade water | Sigma | 270733 |
| Methanol | VWR | BDH20864.400 |
| SpeedVac | Thermo Fisher | SPD120 |
| Vanquish LC | Thermo Scientific | NA |

| Reagent/Resource | Reference or Source | Identifier or Catalog Number |
|---|---|---|
| Exploris 240 | Thermo Scientific | NA |
| Mass Spectrometry Metabolite Library of Standards | IROA Technologies | MSMLS |
| Kingfisher Apex System | Thermo Scientific | 5400910 |
| Easy-nLC 1200 | Thermo Scientific | NA |
| Q-Exactive HF Orbitrap mass spectrometer | Thermo Scientific | NA |
| Empore SPE C18 | The Nest Group Inc. | 1311018 |
| BioPureSPN PROTO 300 C18 Mini or Midi columns | The Nest Group Inc. | NC1678004 |
| 50 cm µPAC nano-LC column | Thermo Scientific | COL-NANO050NEOB |
| iRT peptides | Biognosys | NA |
| **Software** | | |
| GraphPad Prizm 9 | https://www.graphpad.com | v.9 & v.10 |
| Imagej | https://imagej.nih.gov/ij/index.html | 1.54i |
| TraceFinder | Thermo Scientific | versions 5.1 and 5.2, |
| Spectronaut Pulsar X | Biognosys | NA |
| **Other** | | |
| H&E and IHC slide scans | Leica biosystems v.12 | Aperio ImageScope |
| Transcriptomes sequenced | | NextSeq500 using v2 chemistry |
| hatcher incubator | GQF, Berryhill, ON Canada | Digital 1502 sportsman |
| rotary cutting tool | Dremel 3000-N/18, USA | F0133000AW |
| weigh boats | Fisher Scientific | Cat. No. 08-732-113 |
| Motic SMZ-171with cellSens Standard | Olympus Corp., Tokyo, Japan | NA |
| Syringe plunger | BD | #309602 |
| cell strainer 40 µm | Fisher brand | CT#22363547 |

## Ethical regulations

Patient specimens were collected by Biobank staff at BC Children's Hospital. Consenting of the participant and collection of the specimen were conducted following review and approval by the University of British Columbia Children & Women's Research Ethics Board (H17-01860, H18-02473-A037), and conformed with standards defined in the WMA Department of Helsinki and the Department of Health and Human Services Belmont Report. The studies using the chick embryo CAM model followed BCCHR institution internal protocols and University of British Columbia animal committee guidelines. Experiments with the CAM model in this paper adhered to Canadian Council of Animal Care (CCAC) regulations that did not require approval and ARRIVE guidelines 2.0 (Percie du Sert et al, 2020). Additionally, avian embryos are not considered live vertebrate animals under NIH PHS policy. The use

of zebrafish in this study was approved and carried out according to the policies of the University of Ottawa's Animal Care Committee (Protocol #CHEOe-3195), which is governed and certified by the Canadian Council on Animal Care. All mouse procedures and experiments were reviewed and approved by the University of Calgary Animal Care Committee (AC21-0016).

## Tissue processing

Clinical tissues used in this study were received fresh from the operating room within 30 min of removal. Tissues were triaged into either, snap frozen samples using liquid nitrogen cooling or viably cryopreserved using 10% DMSO with fetal bovine serum (FBS). The samples were processed into blocks using 10% neutral-buffered formalin and paraffin embedding. Slides were sectioned at 4 microns for hematoxylin and eosin (H&E) staining. CAM-generated tumoroid tissues were removed, quickly washed in DPBS, fixed in 10% formalin for 24–48 h, and processed for paraffin embedding (FFPE).

## Immunohistochemistry (IHC)

Human SETTLE tissue, Mouse PDX, CAM-PDX SETTLE, and tissue microarray blocks were sectioned at 4 and 5 µm using Thermo Scientific Rotary Microtome HM340E. The slides were dried at 37 °C overnight and briefly baked at 65 °C before sections were deparaffinized in xylene, then rehydration steps using an ethanol gradient of 100%, 95%, 75%, and 50%, followed by a final wash in double distilled water. Hematoxylin and Eosin (H&E) staining and Immunohistochemistry (IHC) were performed as per the manufacturer's instruction (Vector Laboratories, USA). Briefly, a steam-generated heat-induced epitope retrieval (HIER) method was used for antigen retrieval by citrate buffer using a steam cooker with antigen retrieval solution (10 mM Na citrate pH 6 + 0.05% Tween 20). Deparaffinized and hydrated sections were incubated for 10 min at room temperature with 3% $H_2O_2$ to block endogenous peroxidase activity. After a rinse with PBS, the slides were blocked for unspecific binding with protein Avidin solution (goat serum in PBS) for 20 min, at room temperature. Anti-SHMT2, a rabbit polyclonal antibody (Proteintech, USA; Cat No. 11099-1-AP) was added at a concentration of 1 µg/ml and incubated overnight at 4 °C. The biotinylated secondary antibody was added to a blocking buffer and incubated for 30 min at room temperature. Slides were washed with PBS and then the Horse-radish peroxidase (HRP) enzyme conjugated to avidin (ABC reagent) was added for 30 min at room temperature. Similarly, the LAMP1, an anti-human CD107a antibody (clone H4A3, Biolegend, USA) was applied at the concentration of 0.5 mg/mL and incubated overnight at 4 °C, the procedure was followed by PBS washings and by the addition of biotinylated secondary antibody (HRP) streptavidin conjugate (1:200 for 30 min) and finally incubated with ABC reagent for 25–30 min, (VECTASTAIN Elite, Vector Laboratories, USA). An isotype control IgG antibody was maintained against both the primary antibodies in parallel. Slides were washed with PBS and incubated with Liquid DAB+ (3-3'-Diaminobenzidine) substrate-chromogen System (Dako, North America) according to the manufacturer's protocol. After a rinse in water, the slides were counterstained with Hematoxylin, dehydrated, cleared, and mounted with coverslips by ShurMount

and Cytoseal 60 (Epredia). All H&E and IHC slides were scanned with a Leica Aperio AT2 slide scanner and images were processed with Aperio ImageScope (Leica biosystems v.12). Tumor orientation was maintained consistently during the study to observe growth and dissemination during the implantation period.

## Whole genome and transcriptome sequencing, and data analysis

Tumor and normal genomes were sequenced on HiSeqX using v2.5 chemistry. Transcriptomes were sequenced on NextSeq500 using v2 chemistry. Normal and tumor reads were aligned to the human reference genome (hg38) using the Burrows-Wheeler Alignment miniMap2 tool (v2.15). Somatic point mutations (SNVs) and small insertions and deletions (indels) were detected using Strelka (v2.9.10) and Mutect2 (v2.4.0). Somatic copy number alterations were identified using CNAseq (v0.0.6), and loss of heterozygosity used APOLLOH (v0.1.1). Structural variants (SVs) in RNA-Seq data were identified using the assembly-based tools ABySS (v1.3.4) and TransABySS (v1.4.10) and alignment-based tools Chimerascan (v0.4.5) and DeFuse10 (v0.6.2); SVs in the DNA sequence data were identified using assembly-based tools ABySS and Trans ABySS and alignment-based tools Manta v1.0.0 and Delly v0.7.3. Putative SV calls identified from the DNA and RNA sequences were merged into a consensus caller MAVIS (v2.1.1).

RNA-seq reads were aligned using STAR (v.2.5.2b) to the human reference (hg38) and expression was quantified using RSEM (v.1.3.0). Publicly available transcriptome sequencing data from Illumina BodyMap 2.0, the Genotype-Tissue Expression (GTEx) Project, The Cancer Genome Atlas, Treehouse Childhood Cancer Initiative, and the TARGET program were used to explore the expression profiles of human genes and transcripts.

## Automated sonication-free acid-assisted proteome (ASAP) tissue lysis workflow

Primary pretreatment resection (Primary), first (R1), second (R2) and third (R3) relapse were utilized for the proteomics analysis. FFPE specimens were fixed in 10% neutral-buffered formalin, 10 μm sections from FFPE blocks were mounted on positively charged glass slides and dried at 37 °C overnight. After deparaffinization and H&E staining, the slides were macrodissected into the tumor (including spindle and epithelial elements) and adjacent normal tissue by a pathologist. The adjacent normal tissue for metastatic lesions captured lung parenchyma including predominantly alveolar epithelium, vascular structures, and a small amount of pleural mesothelium and associated fibroblasts. Small areas of respiratory bronchiole epithelium, smooth muscle, and lymphoid tissue were also included. Red blood cells were present, as were focal areas of alveolar macrophages. The normal tissues were not selected to have minimal non-epithelium elements samples. The adjacent normal for the primary tumor encompassed thyroid tissue including thyroid follicles, stromal fibroblasts and vascular structures but excluding adjacent skeletal muscle.

Each timepoint was processed in triplicate, where each triplicate consisted of three 10 μm thick tissue sections. The macro dissected H&E-stained tissues were processed as described in Barnabas et al (2023). Briefly, samples were resuspended in 25 μl 75% trifluoroacetic acid (TFA) for 10 min at room temperature. 175 μl

Neutralization buffer (2 M Tris, 3% SDS) was added and incubated at 95 °C for 80 min followed by reduction and alkylation with 10 mM TCEP and 40 mM CAA at 95 °C for 20 min.

Proteins were extracted by SP3 as published (Hughes et al, 2019) using the Thermo KingFisher Apex platform. 10 μl hydrophobic SpeedBead Magnetic Carboxylate bead (stock solution 50 mg/ml; G&E 651521050250) and 10 μl hydrophilic SpeedBead Magnetic Carboxylate bead (stock solution 50 mg/ml; G&E 451521050250) were mixed per sample and bound to a magnet for 1 min. The beads were washed 3 times with 1 ml of MS-grade $H_2O$ and protein binding was induced by adding a final concentration of 80% Ethanol for 10 min. The SP3 beads were washed three times with 90% Ethanol followed by on-bead protein digestion in 80 μl 100 mM Ammonium bicarbonate containing 0.5 μg Trypsin/LysC Mix (Promega V507A) (enzyme/protein ratio of 1:50). The digestion mixture was incubated for 16–18 h at 37 °C. Peptides were eluted from the SP3 beads using the KingFisher Apex robot and the peptide concentration was measured using the Pierce Quantitative Colorimetric Peptide assay according to the manufacturer's protocol. Peptides were purified on Empore SPE C18 STAGE tips (Rappsilber et al, 2007) or BioPureSPN PROTO 300 C18 Mini or Midi columns (The Nest Group Inc.) according to the manufacturer's protocol. Purified peptides were concentrated using a vacuum concentrator and resuspended in 0.1% Formic Acid (FA) containing iRT peptides at 1:300 final dilution (Biognosys).

## Liquid chromatography and mass spectrometry

1 μg peptide per sample were injected into an Easy-nLC 1200 liquid chromatography system coupled to a Q-Exactive HF Orbitrap mass spectrometer (Thermo Scientific). Buffer A contained 2% acetonitrile (ACN) and 0.1% FA and Buffer B was 95% ACN and 0.1% FA. The peptides were separated on a 50 cm μPAC nano-LC column (reverse phase C18, pore size 100–200 Å, Thermo Scientific) with a flow rate of 300 nL/min and a gradient of 2 to 26% buffer B over a 180-min gradient. A full MS scan with a mass range of 300–1650 $m/z$ was collected with a resolution of 120,000. Maximum injection time was 60 ms and AGC target value was $3 \times 10^6$. DIA segment MS/MS spectra were acquired with a 24-variable window format with a resolution of 30,000. AGC's target value was $3 \times 10^6$. Maximum injection time was set to 'auto'. The stepped collision energy was set to 25.5, 27.0, and 30.0.

De-salted peptides from samples were combined into a single pool and analyzed in ten gas-phase fractions by injecting 1 μg per in a 3 h gradient. The first eight fractions (340–820 $m/z$) were analyzed over a 60 $m/z$ window (e.g., fraction 1: 340–400 $m/z$), each with a loop count of 30 and a window size of 2 $m/z$. The last two fractions (820–1180 $m/z$) were analyzed over a 180 $m/z$ window each, with a loop count of 30 and 6 $m/z$ window.

Proteomic raw data was searched using Spectronaut Pulsar X (Biognosys) using a human FASTA from UniProt (Dec 2021) and a FASTA file for iRT peptides (Biognosys) was included in the search. For the search, enzyme, and digestion type were set to Specific, and Trypsin/P, acetyl (Protein N-term), and oxidation (M) were set as variable modifications and carbamidomethyl (C) was set as fixed modification. Maximum and minimum peptide length were set to 7 and 52 amino acids, respectively, and missed cleavage was set to 2. Precursor and protein FDR were set to 1% and a minimum of 2 peptides were used for quantification. The data was normalized in

Spectronaut based on precursors identified in 70% of the samples. Data was filtered for proteins identified in 70% of tumors or adjacent normals and missing values were imputed using a down-shifted normal imputation strategy in perseus (Tyanova et al, 2016). Student's t-test was performed using perseus and proteins with log2 fold change >1 with an FDR threshold of 5% was filtered for enrichment analysis. Gene ontology enrichment analysis was performed using Gorilla (Eden et al, 2009). We utilized the pyCirclize package from https://github.com/moshi4/pyCirclize for the circular plot. The log2 fold-change values, calculated from tumor vs normal comparison, are displayed on the proteomics side of the plot. The genes depicted are those targeted by at least one of the commonly used drugs in pediatric cancer. For the dotplot, cancer versus normal testing is performed using independent sample t-tests with Benjamini-Hochberg correction. The $p$-value is as follows: not significant: $p \leq 1.00e + 00$, *: $1.00e-02 < p \leq 5.00e-02$, **: $1.00e-03 < p \leq 1.00e-02$, ***: $1.00e-04 < p \leq 1.00e-03$, ****: $p \leq 1.00e-04$.

## Establishment of SETTLE PDX in CAM

Fresh tumor tissue samples were collected at the time of surgery by the PROFYLE West biobank at British Columbia Children's Hospital (BCCH), submerged in viable cryopreservation media (10% DMSO in 10% FBS/RPMI), rate control cooled ($-1\,°C$ per min from $4\,°C$ to $-80\,°C$), and stored in vapor phase liquid nitrogen.

A portion of the cryopreserved human SETTLE tumor sample was used to establish xenografts in chick chorioallantoic membrane (CAM). Fertilized eggs from White Leghorn chickens (Gallus gallus domesticus) were purchased from the University of Alberta, Edmonton, Alberta. Following overnight reset of the air sac at $14\,°C$, eggs were placed into a hatcher incubator (Digital 1502 sportsman, GQF, Berryhill, ON Canada) at $37\,°C$ with relative humidity >70% for 4 days. The *ex-ovo* method for tumor engraftment was used for all CAM-PDX studies (Javed et al, 2022). Eggs were cracked aided with a rotary cutting tool (Dremel) and the deshelled CAM was placed in lidded sterile weigh boats (Cat. No. 08-732-113, Fisher Scientific). CAMs were further incubated at $37\,°C$ with humidity >70% until embryonic day 11 (ED11), when tumors are engrafted.

The viably cryopreserved SETTLE tissues were thawed rapidly, rinsed with Dulbecco's phosphate-buffered saline (DPBS, Gibco, Waltham, MA) and submerged in fetal bovine serum (FBS, Gibco, Waltham, MA). Tumor fragments were cut into smaller 1–2 mm pieces for implantation. Tumor fragments were engrafted onto gently lacerated membrane of ED11 CAMs, and immediately overlaid with 15–20 µL of cold Geltrex (Lot#A1413.02 & A14132-02, Gibco, NY USA). Tumor grafted CAMs (herein referred to as tumoroids) were incubated at $37\,°C$ with humidity for up to 4–7 days, depending on the experimental workflow with routine imaging conducted to assess tumoroid growth. Tumors were typically harvested between D4-D7 following implantation for cryopreservation in 10%DMSO with RPMI 1640 media (Gibco, Waltham, MA), or formalin-fixed for paraffin embedding (FFPE) and histological workups.

## Targeted drug response of SETTLE in CAM-PDX

Prior to drug treatments in CAM-PDXs, preliminary safety studies were conducted for each drug to assess the maximum tolerated dose in non-tumor-bearing CAMs. For primary SETTLE CAM-PDXs, sertraline (S4053, Selleckchem, USA), SHIN1 (#6998, Tocris), or vehicle control were administered directly onto the tumoroids, on the day of implantation (D1) admixed with Geltrex, and with topical redosing at D3. Treatment response was assessed by imaging the CAM-PDX tumoroids at regular intervals (D1, D3, D5) (Motic SMZ-171 stereo microscope equipped with an Olympus DP72 color camera and cellSens Standard (Olympus Corp., Tokyo, Japan). The tumoroid periphery/area was measured using ImageJ software.

A modified protocol was used to assess treatment response to sertraline and/or trimethoprim of mouse-expanded SETTLE tumors in CAM-PDXs (primary non-expanded tumors had been exhausted). Viably cryopreserved mouse PDX tissue pieces were rapidly thawed, washed in DPBS, and processed by scalpel-based cutting into numerous small pieces. The tumor pieces were then pressed gently with a syringe plunger (#309602, BD) to dissociate the tissue into a homogeneous cell suspension that was passed through a cell strainer (40 µm, CT#22363547, Fisher brand). The cells were pelleted by centrifugation, and the cell pellet resuspended in RPMI media in sufficient volume for the requisite group numbers in the study. To prepare gel spheroids for implantation, the tumor cell suspension was mixed with an equal volume of Geltrex and pipetted onto a pre-warmed petri dish using the hanging drop method to form dome-shaped spheroids. After allowing the spheroids to set at $37\,°C$ for 30 min, complete media was added and further incubated for 2 h until onplantation on lacerated CAMs. Tumoroids received topically administered treatments of vehicle only control, sertraline and/or trimethoprim at the indicated concentrations at the onset of implantation (D1) and re-applied on day 3 (D3).

## Sertraline treatment of MDA-MB-468 and MDA-MB-231 cells as CAM xenografts

MDA-MB-231 and MDA-MB-468 (triple negative breast cancer) cell lines (STR authenticated and mycoplasma free) were cultivated in compete DMEM media (10% FBS, pen/strep). On day of CAM engraftment, cells were harvested by trypsin/EDTA, neutralized and washed, and resuspended in DMEM. Briefly, $1 \times 10^6$ cells was mixed with an equal volume of Geltrex and pipetted onto a pre-warmed petri dish using the hanging drop method to form dome-shaped spheroids and allowed to set at $37\,°C$ for 30 mins. Following which, complete media was added and further incubated for 4 h until engraftment on lacerated ED11 CAMs. Tumoroids received topically administered treatments of vehicle only control or sertraline at the indicated concentrations at the onset of implantation (D1) and re-applied on day 3 (D3).

## Establishment of SETTLE PDX in NSG mice

A portion of the cryopreserved human SETTLE tumor sample was used for the establishment of a patient-derived murine xenograft, under institutional review board (IRB) approval (HREBA.CC-16-0144). Six- to eight-week-old female NOD.Cg-Prkdcscid Il2rgtm1Wjl/SzJ (NSG) mice (Jackson Laboratories, Bar Harbor, MA, USA) were housed in groups of five and maintained as the procedures reviewed and approved by the University of Calgary Animal Care Committee (AC21-0016). Briefly, NSG mice were anesthetized via intraperitoneal injection of 50 mg/kg of ketamine

and 5 mg/kg of xylazine. A 5 mm section of tumor tissue was implanted subcutaneously through an incision on the right hind flank. Following surgery, mice were administered two doses of buprenorphine (0.05 mg/kg, subcutaneously), 12 h apart, and monitored three times per week for signs of tumor growth. Once tumors had established (6–8 months), the mouse was sacrificed (intraperitoneal injection of 100 mg/kg ketamine and 10 mg/kg xylazine followed by cervical dislocation), and the tumor was collected (in vivo passage 1). Tumor tissue was then implanted into 5 mice (as described above) collected and stored. The presence of human tumor cells in PDX-generated samples was confirmed by immunohistochemistry using human-specific nucleolin antibody.

To prepare dissociated SETTLE PDX tumor cells, surgically removed NSG-PDX tissue was rinsed in sterile saline and placed on ice. Tissue was cut into 5 mm pieces and enzymatically dissociated using the Miltenyi Human Tumor Dissociation Kit and the Miltenyi GentleMACS Octo (with heaters) dissociator system (program: 37C_h_TDK_2). Following dissociation, the sample was filtered through a 70 μm filter, cells were counted, centrifuged at $300 \times g$ for 7 min, supernatant discarded, and the cell pellet subjected to the Miltenyi Mouse Cell Depletion Kit (Miltenyi Biotec). The purified human cells were then aliquoted into 2 mL cryovials at a density of 3–5 million cells in 1 mL of Cryostore CS10 and stored as described above.

## In vitro targeted drug response of SETTLE PDX cells

Targeted drug response was assessed in vitro using the aforementioned dissociated and cryopreserved SETTLE PDX cells (from mouse-depleted PDX tumors). Cells were thawed rapidly in a 37 °C water bath, transferred to 10 mL CoDMEM (DMEM without pyruvate (Corning), with 10% FBS (Invitrogen) and penicillin-streptomycin (Invitrogen)), spun down, and resuspended either in fresh CoDMEM with 10% FBS or 10% dialyzed FBS. Where indicated, SETTLE cells were cultured in standard tissue culture dishes for up to 7 days in CoDMEM prior to drug assays. For comparative purposes, some experiments utilized MDA-MB-231 and MDA-MB-468 cell lines (ATCC HTB-26 and HTB-132) maintained at 37 °C, 5% $CO_2$ in cDMEM (DMEM (Sigma-Aldrich), with 10% fetal bovine serum (FBS, Invitrogen), penicillin-streptomycin (Invitrogen), and non-essential amino acids (Invitrogen)), and switched to CoDMEM for the drug assays.

Cells were lifted using trypsin-EDTA, washed and resuspended in media. 5000 live cells were plated per well of a 96-well plate and allowed to adhere overnight. Next day, the media was drained and replaced with drug (sertraline, artemether and/or trimethoprim) containing media (pre-diluted at the indicated concentrations) and incubated for 4 days. Cell viability was assessed with CellTiter-Glo Luminescent Cell Viability Assay (Promega), according to manufacturer's instructions. Prism (GraphPad) was used to generate dose response plots and to calculate IC50 values and statistics. Additional plots and calculations of combinatorial drug synergies using the HSA (Highest Single Agent) reference model utilized SynergyFinder3.0 (Ianevski et al, 2022).

## Drug treatment response of mouse-derived cells in larval zebrafish PDX

Adult Casper zebrafish (White et al, 2008) were bred in accordance with standard protocols (Westerfield, 2000) and maintained under controlled conditions (28 °C, 14 h:10 h light: dark cycle) in the aquatics facility at the University of Ottawa. All experimental procedures were conducted in compliance with the University of Ottawa's Animal Care Committee policies (Protocol #CHEOe-3195), which adhere to the guidelines of the Canadian Council on Animal Care. The study was conducted in compliance with animal care committee policies and guidelines. To assess the toxicity of drugs of interest, in vivo studies were conducted on Casper zebrafish larvae using immersion therapy. Cryopreserved dissociated SETTLE cells (mouse expanded) were thawed and processed using a dead cell removal kit (Miltenyi Biotec, #130-090-101) to obtain viable single cells, which were then labeled with CellTracker DeepRed cytoplasmic fluorescent dye (ThermoFisher, #C34565). The labeled cells were resuspended in Dulbecco's Modified Eagle Medium (DMEM) supplemented with 10% heat-inactivated fetal bovine serum (FBS) for injection into 48-h post-fertilization (hpf) zebrafish larvae. Larvae were anesthetized with tricaine (0.3 mg/mL) and arrayed in agarose injection plates for cell transplantation. Using a pulled capillary needle, 100–200 cells were manually injected into the yolk sac (YS) of each larva, followed by incubation at 35 °C. One day post-injection (1 dpi), larvae were screened under a Far-Red filter (620–700 nm) to confirm the presence of human tumor cells in the YS. Larvae with tumor engraftment were divided into groups and treated with solvent control, single agents, or drug combinations via immersion therapy for 72 h (about 3 days). To evaluate drug response, ex vivo tumor cell quantification was performed at 1 dpi (baseline/untreated larvae) and 4 dpi (endpoint). Larvae were dissociated in a collagenase solution, and the resulting cell suspension was centrifuged and washed before resuspending in 10 μL per embryo for imaging. Post-quantification, fluorescent tumor cell numbers were assessed using FIJI software.

## Data analysis

All CAM experimental data were analyzed using GraphPad Prism10.0 software. Results were expressed as the Mean ± SD. Statistical tests are as indicated in the figure legends. Values represented as non-significant (ns) ≥0.05, *<0.05, **<0.01, ***<0.001, ****<0.0001. Blinding was not performed during study group assignment, data collection and primary analysis. Blinding was conducted during secondary analysis to remove knowledge bias of drug-treated vs non-treated groups.

## Metabolic tracing

A vial of cryopreserved SETTLE PDX cells (from mouse depleted PDX tumors) was quickly thawed, resuspended in pre-warmed stable-isotope tracing media, centrifuged at $300 \times g$ for 5 min to remove cryopreservation media, resuspended in tracing media, and plated equally onto 6-well plates containing either vehicle (DMSO) or 10 μM sertraline. Stable-isotope tracing media was formulated by supplementing custom DMEM lacking glucose and amino acids (US Biological, D9800-27) with 10% dialyzed fetal bovine serum and $^{13}C_6$-glucose/unlabeled glucose, $^{13}C_3$-serine/unlabeled serine, and other unlabeled amino acids; as indicated. Following the indicated culture period, cells were quenched by quickly aspirating media and washing with ice cold 0.9% NaCl, prepared in HPLC-grade water (Sigma, 270733), before adding 1 mL of ice-cold extraction buffer. The extraction buffer consisted of 80% methanol

(VWR, BDH20864.400), 20% water (Sigma, 270733). Cells were scraped into extraction buffer, transferred to tubes, vortexed at 4 °C for 5–10 min, and centrifuged at 4 °C at maximum speed for 15 min. 0.9 mL of supernatant was transferred to a new tube and concentrated using a SpeedVac (Thermo Fisher, SPD120) until dry. Metabolites were reconstituted into 25–50 µl of water, vortexed, centrifuged, and transferred to vials for analysis by LCMS.

Samples were loaded into a temperature controlled (6 °C) autosampler and subjected to an LCMS analysis to detect and quantify peaks corresponding to annotated metabolites. A ZIC-pHILIC (2.1 × 150 mm, 5 µm; Millipore) LC column was coupled to a Vanquish LC (Thermo Scientific). The volume oven temperature was set to 25 °C, maintained by forced air and an integrated column heater. The column was pre-equilibrated using a flow rate of 100 µl/min and 80–20%B. Following injection of 2 µl of sample, the following gradient elution at 100 µl/min was used: 80–20%B (0–30 min), 20–20%B (30–40 min), and 20–80%B (40–40.5 min); the LC column was re-equilibrated using 80–80%B from 40.5–52 min before subsequent injections. Mobile phase composition was: (A) 10 mM ammonium carbonate in HPLC-grade water, pH 9.0 and (B) acetonitrile, 100%. Mobile phase A was freshly prepared or used within one week.

The LC was coupled to an Exploris 240 (Thermo Scientific) mass spectrometer operating in heated electrospray ionization mode (HESI) for analysis. The following parameters were set for HESI: spray voltage 3.4 kV (positive) and 2 kV (negative), static spray voltage, sheath gas 25, aux gas 5, sweep gas 0.5, ion transfer tube temperature 320 °C, and vaporizer temperature 75 °C. The global parameters included an expected peak width of 20 s, mild trapping, and a default charge state of 1. A 40-min polarity switching data-dependent Top 5 method was used for positive mode and a data-dependent Top 3 method was used for negative mode. Full MS scan parameters for both positive and negative modes were set as follows: scan range 67–1000 $m/z$ collected in profile mode, Orbitrap resolution 120,000, RF lens 70%, AGC target of 300%, and maximum injection time set to automatic. ddMS2 for positive mode were collected in centroid mode at an Orbitrap resolution of 30,000, isolation window of 1.5 $m/z$, an AGC target set to standard, a maximum injection time set to automatic, and a normalized collision energy set to 10%, 30%, and 80%. ddMS2 for negative mode were collected in centroid mode at an Orbitrap resolution of 30,000, isolation window of 2 $m/z$, an AGC target set to standard, a maximum injection time set to automatic, and a normalized collision energy set to 30%. For both positive and negative ddMS2, we applied an intensity threshold of 5e4 and a dynamic exclusion of 5 ppm for 10 s, excluding isotopes.

Raw LCMS data were analyzed using TraceFinder (versions 5.1 and 5.2, Thermo Scientific). Peaks corresponding to specific metabolites were annotated using a retention time and accurate mass library created from the Mass Spectrometry Metabolite Library of Standards (MSMLS, IROA Technologies) and other authentic standards acquired from Sigma. Peak areas were used to quantify the relative abundance of each metabolite and were quantified using the following general ICIS algorithm settings: a $m/z$ discrimination threshold of 5 ppm, a retention time window of 60 or 120 s, a minimum peak area of 5e5, a smoothing factor of 9, an area noise factor of 5, a peak noise factor of 10, a baseline window of 100, a minimum peak height signal-to-noise ratio of 2, a minimum peak width of 5, a multiplet resolution of 10, an area tail extension of 20, and an area scan window of 0. Each peak

**The paper explained**

**Problem**

Despite significant advances in diagnosis and treatment, cancer continues to be a leading cause of death among children, adolescents, and young adults. Genomics has reshaped the diagnostic landscape of pediatric cancers by identifying and incorporating actionable genetic features that enhance diagnosis, classification, and treatment strategies. However, translating these precision oncology insights into effective therapies for difficult-to-treat childhood, adolescent, and young adult cancers remains a substantial challenge.

**Results**

We integrate proteomics with patient-derived xenograft models to identify a personalized treatment for an adolescent with primary and metastatic spindle epithelial tumor with thymus-like elements (SET-TLE). Proteomics identified elevated SHMT2, a key regulator of one-carbon metabolism, as a potential therapeutic target, leading to treatment with the anti-depressant sertraline. The drug response was validated using a personalized chicken embryo and zebrafish xenograft models of the patient's SETTLE tumor. After the failure of cytotoxic chemotherapy and second-line therapies, the patient was treated with sertraline, resulting in a slower tumor growth rate, though the disease remained clinically progressive.

**Impact**

Our study highlights the value of combining proteomics and personalized xenograft models with genomics to improve precision oncology for rare pediatric cancers. It demonstrates how this integrated approach can guide personalized treatment decisions, accelerate therapeutic evaluations, and provide clinically relevant data, potentially transforming medical decision-making and clinical practice.

integration was manually inspected, and individual settings were adjusted to optimize integration.

## Data availability

To maintain the patient's and family's privacy in this study of a single case the raw data is not made available at this point. Release of the raw data as part of an aggregated patient cohort is in preparation. Please contact the corresponding author(s) to discuss individual access to de-identified raw data through secure systems.

The source data of this paper are collected in the following database record: biostudies:S-SCDT-10_1038-S44321-025-00212-8.

## Peer review information

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

## Acknowledgements

This work was supported by the PRecision Oncology For Young peopLE (PROFYLE) program and the BC Children's Hospital Foundation through the Better Responses through Avatars and Evidence (BRAvE) Initiative. Salary support was provided by the Michael Cuccione Foundation (CJL, GSDR, CAM, PFL, and VG), the Canada Research Chairs Program (CRC-RS 950-230867, PFL), the Canadian Institutes of Health Research (CAM and PFL), the Michael Smith Foundation for Health Research Scholar Program (16442, PFL), MITACS (TAB and GB) and the University of British Columbia (EKE). Project support for JAC and DLS was provided by The Alberta Cancer Foundation. We thank all the authors for their contributions and technical assistance. This manuscript was edited at Life Science Editors. We are indebted to Dr Karla Williams (UBC Pharmaceutical Sciences) for vital assistance in establishing the CAM facility at BCCHRI. We gratefully acknowledge the participation of the patients and families that made this study possible and the BC Children's Hospital staff physicians and Biobank staff for their tremendous efforts in collecting and maintaining specimens.

## Author contributions

**Georgina D Barnabas**: Conceptualization; Data curation; Formal analysis; Investigation; Visualization; Methodology; Writing—original draft; Writing—review and editing. **Tariq A Bhat**: Conceptualization; Data curation; Formal analysis; Investigation; Visualization; Methodology; Writing—original draft; Writing—review and editing. **Verena Goebeler**: Investigation; Visualization; Writing—review and editing. **Pascal Leclair**: Investigation; Visualization; Writing—review and editing. **Nadine Azzam**: Investigation; Visualization; Writing—review and editing. **Nicole Melong**: Investigation; Visualization. **Colleen Anderson**: Investigation. **Alexis Gom**: Investigation. **Seohee An**: Investigation. **Enes K Ergin**: Investigation. **Yaoqing Shen**: Formal analysis; Investigation; Writing—review and editing. **Agustina Conrrero**: Data curation.; Visualization **Andrew J Mungall**: Investigation. **Karen L Mungall**: Investigation. **Christopher A Maxwell**: Funding acquisition; Writing—review and editing. **Gregor S D Reid**: Supervision; Funding acquisition; Writing—review and editing. **Martin Hirst**: Investigation. **Steven Jones**: Investigation; Writing—review and editing. **Jennifer A Chan**: Investigation; Methodology; Writing—review and

editing. **Donna L Senger**: Supervision; Funding acquisition; Investigation; Visualization; Methodology; Writing—review and editing. **Jason N Berman**: Supervision; Funding acquisition; Investigation; Visualization; Methodology; Writing—review and editing. **Seth J Parker**: Investigation; Visualization; Methodology; Writing—review and editing. **Jonathan W Bush**: Data curation; Investigation; Visualization; Writing—review and editing. **Caron Strahlendorf**: Funding acquisition; Writing—review and editing. **Rebecca J Deyell**: Funding acquisition; Investigation; Writing—review and editing. **C James Lim**: Conceptualization; Resources; Data curation; Supervision; Funding acquisition; Investigation; Visualization; Methodology; Writing—original draft; Project administration; Writing—review and editing. **Philipp F Lange**: Conceptualization; Resources; Data curation; Formal analysis; Supervision; Funding acquisition; Investigation; Visualization; Methodology; Writing—original draft; Writing—review and editing; Project administration.

Source data underlying figure panels in this paper may have individual authorship assigned. Where available, figure panel/source data authorship is listed in the following database record: biostudies:S-SCDT-10_1038-S44321-025-00212-8.

## Disclosure and competing interests statement

The authors declare no competing interests.

# Expanded View Figures

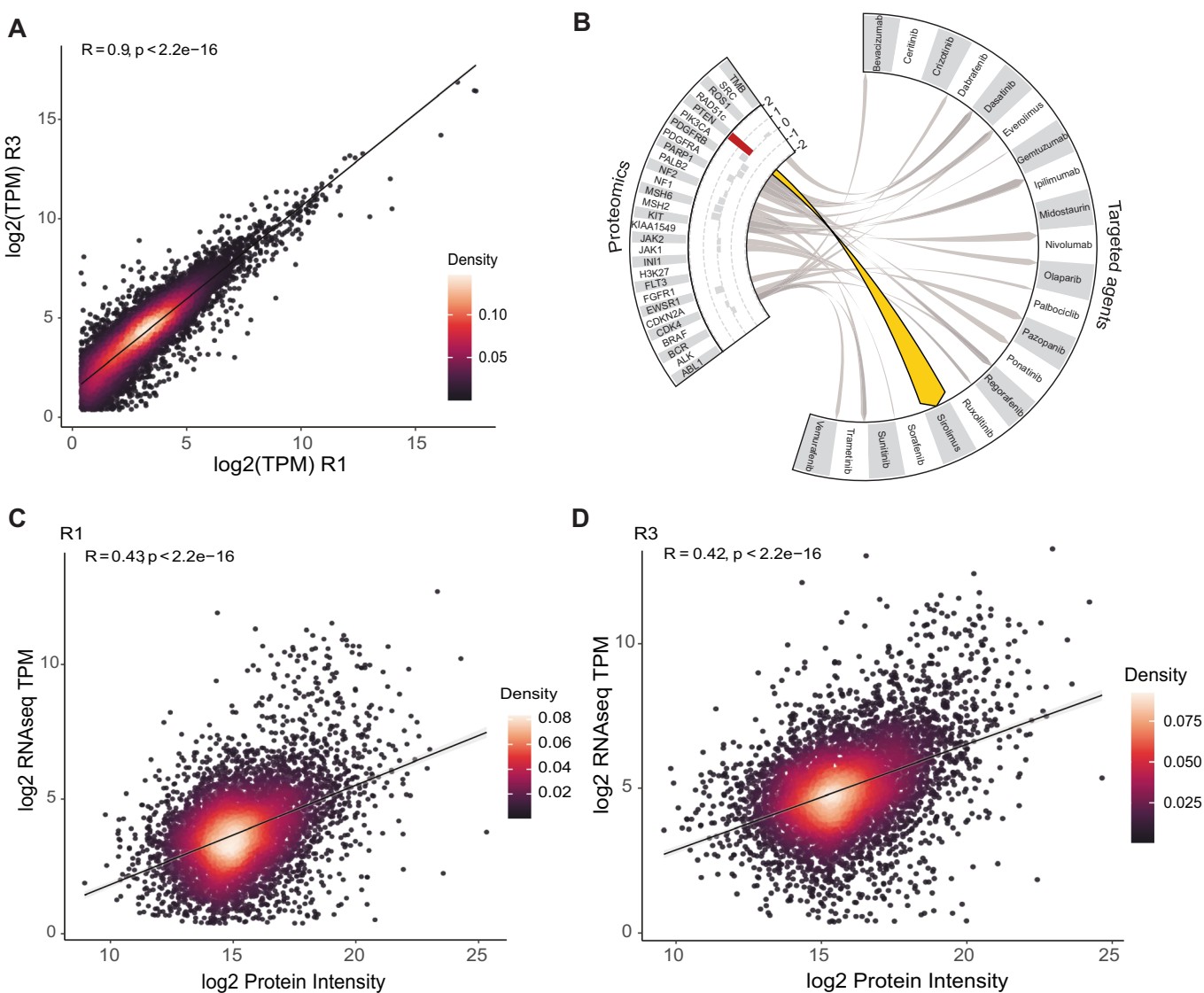

**Figure EV1. Gene and protein expression in progressive SETTLE tumor.**

(A) Correlation plot of gene expression from R1 and R3 with Pearson correlation coefficient r = 0.9, $p = 1e^{-1022}$. (B) Proteome changes do not suggest sensitivity to 21 established therapies. Circular plot showing the targeted proteomic analysis focusing on 29 proteins associated with 21 routinely used therapies in R3 lung nodules. Log2 fold-change values, calculated from the tumor vs normal comparison, are displayed on the proteomics side of the plot. Only associations for quantified proteins are shown. gray arrow: non-significant changes and associations; red arrow: proteome change counter indicative of drug sensitivity. (C, D) Scatter plot comparing protein intensities with gene expression in relapse R1 (C), with Pearson correlation coefficient r = 0.43, $p = 4.4e^{-261}$ and R3 (D), with Pearson correlation coefficient r = 0.42, $p = 2.4e^{-254}$.

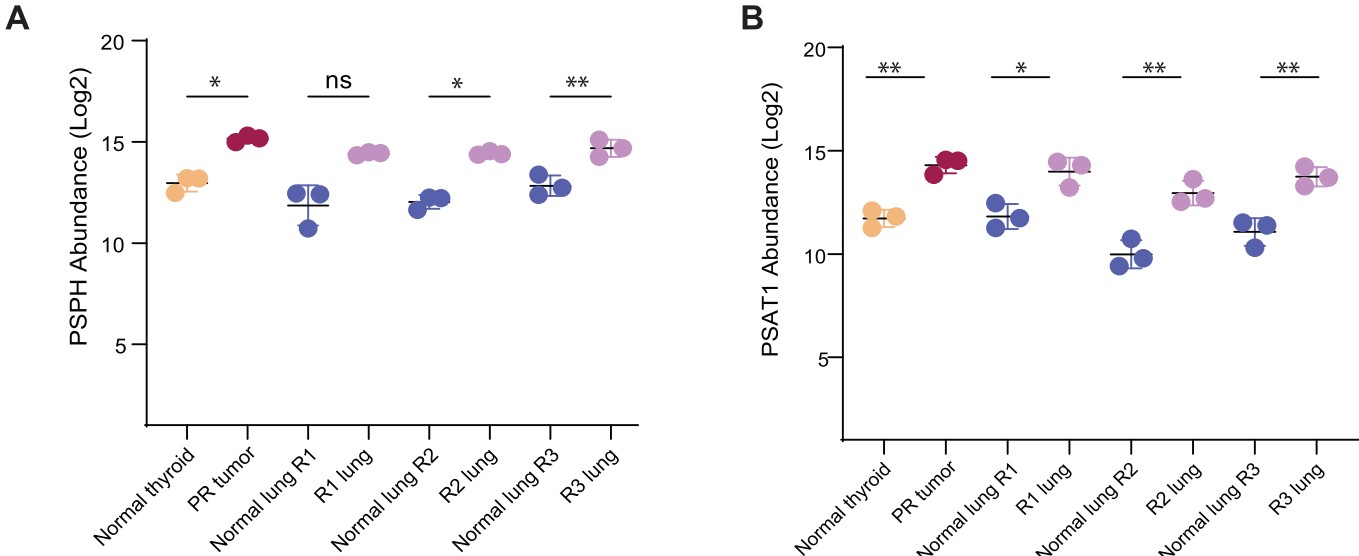

**Figure EV2.  Abundance of serine biosynthesis proteins in SETTLE.**

(A, B) Increased abundance of serine biosynthesis proteins PSPH (A) and PSAT1 (B) in the primary tumor and lung metastases. Data shown represent three technical replicates for each group (mean ± s.d.), Bonferroni's multiple comparisons test, for PSPH: normal thyroid vs PR tumor *p(adjusted) = 0.0208, normal lung R1 vs R1 lung [ns]p(adjusted) = 0.1596, normal lung R2 vs R2 lung *p(adjusted) = 0.0201, normal lung R3 vs R3 lung **p(adjusted) = 0.0054; for PSAT: normal thyroid vs PR tumor **p(adjusted) = 0.0015, normal lung R1 vs R1 lung *p(adjusted) = 0.0142, normal lung R2 vs R2 lung **p(adjusted) = 0.0046, normal lung R3 vs R3 lung **p(adjusted) = 0.0048. Source data are available online for this figure.

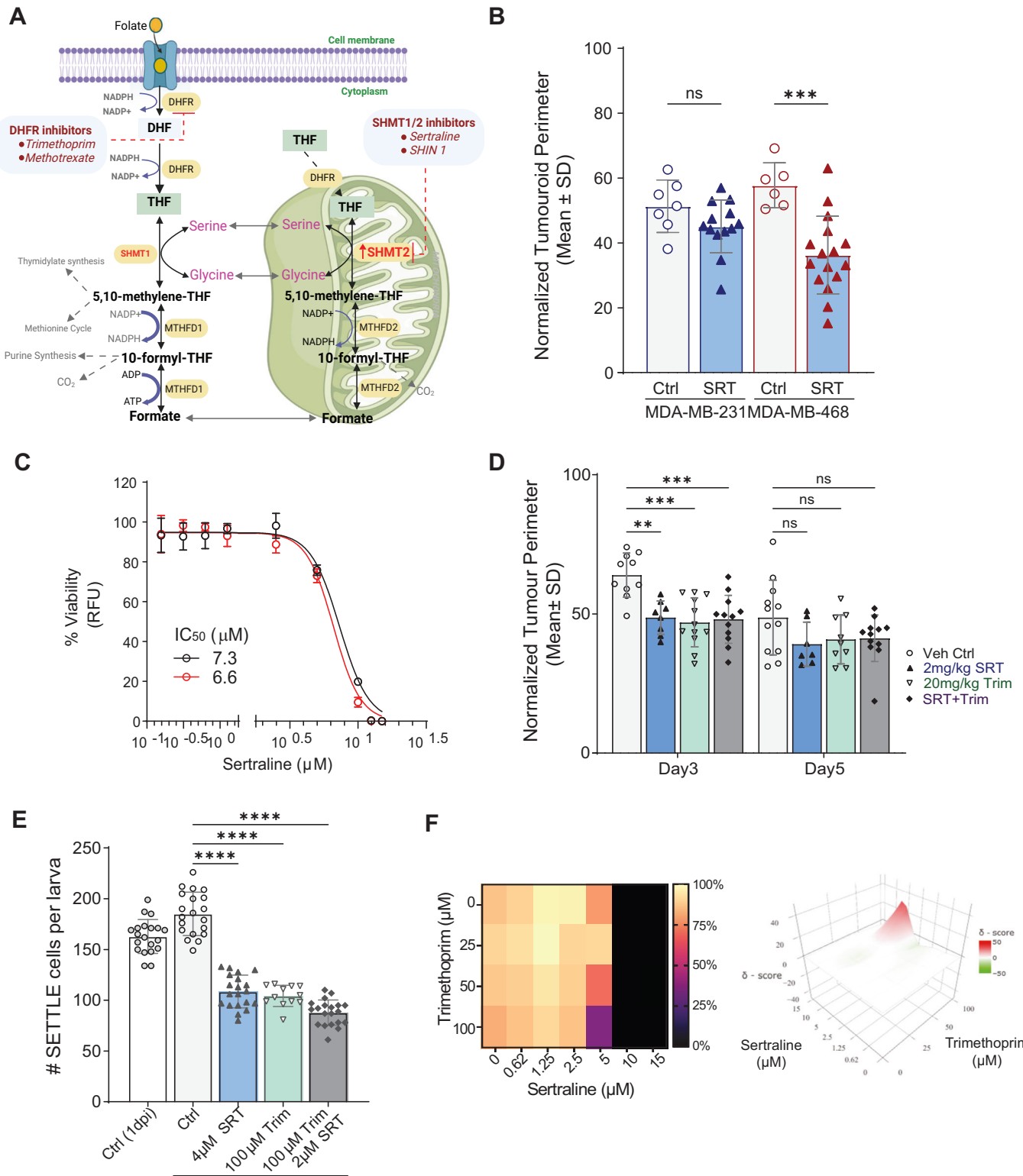

◀ **Figure EV3. Sertraline and trimethoprim combination as a potential therapeutic approach for SETTLE.**

(A) Schematic representation of the 1C metabolism pathway showing the significance of therapeutic inhibition of this pathway at DHFR and SHMT2 levels. Serine is converted to glycine by cytoplasmic SHMT1 (left) and mitochondrial SHMT2 (right). The 1C component sliced from serine is transferred to THF, generating methylene-THF. Therapeutic interventions, highlighted in red-dash lines (DHFR inhibitor: Trimethoprim, SHMT2 inhibitor: Sertraline), at two distant ends of this pathway stop the 1C unit being used for THF. THF is produced from folate and serves as a universal 1C acceptor. *DHF: dihydrofolate; THF: tetrahydrofolate; DHFR: dihydrofolate reductase; MFT: mitochondrial folate transporter; SHTMT1/2, serine hydroxymethyl transferase, cytosolic (1)/mitochondrial (2); MTHFD1: methylenetetrahydrofolate dehydrogenase 1; MTHFD2: methylenetetrahydrofolate dehydrogenase 2 (2-like).* (B) Tumoroid perimeters at Day 4, normalized to Day 1, for CAM xenografts of MDA-MB-468 or MDA-MB-231 cells with and without treatment with sertraline (mean ± s.d., $n = 7;13$ tumors/group for MDA-MB-231, $n = 6;16$ tumors/group for MDA-MB-468, One-way ANOVA with Tukey's post hoc test ***$p = 0.0002$). (C) In vitro viability assays of NSG-PDX SETTLE cells subjected to treatment with sertraline. As plotted is the mean±s.d. for triplicate wells for each concentration of sertraline, for 2 independently conducted assays. (D) CAM engrafted with NSG-PDX SETTLE cells were untreated or treated with sertraline and/or trimethoprim at the indicated concentrations. The scatter bar graph depicts the tumoroid perimeters at days 3 and 5 normalized against day 1 (mean ± s.d., $n = 7$–12 tumors/group, two-way ANOVA with Dunnett's post hoc test, Veh Ctrl vs 2 mg/kg SRT **$p = 0.0041$, Veh Ctrl vs 20 mg/kg Trim ***$p = 0.0074$, Veh Ctrl vs SRT +Trim ***$p = 0.0074$). (E) NSG-PDX SETTLE cells engrafted in larval zebrafish were untreated or treated with sertraline (Sert) and/or trimethoprim (Trim). Scatter bar graphs depict the number of SETTLE cells per larva at 1 or 4 days post-implantation (dpi) (mean ± s.d., n(Ctrl) = 20; n(SRT) = 20; n(Trim) = 12; n(Trim+SRT) = 19 larval zebrafish/group, One-way ANOVA with Dunnet's post hoc test; ctrl vs 4 μM SRT ****$p = 0.0001$, ctrl vs 100 μM Trim ****$p = 0.0001$, ctrl vs 100 μM Trim+2μM SRT ****$p = 0.0001$). (F) In vitro viability assays of NSG-PDX SETTLE cells subjected to treatment with sertraline and/or trimethoprim shown as heat map viability plots (left) and the Highest Single Agent combinatorial drug synergy plots (d-score >10 indicative of synergistic effects) (right). Source data are available online for this figure.

