## [Peer Review File · EMBO Molecular Medicine]

Proteomics and personalized PDX models identify treatment for a progressive malignancy.

Georgina Barnabas, Tariq Bhat, Verena Goebeler, Pascal Leclair, Nadine Azzam, Nicole Melong, Colleen Anderson, Alexis Gom, Seohee An, Enes Ergin, Yaoqing Shen, Agustina Conrroero, Andrew Mungall, Karen Mungall, Christopher Maxwell, Gregor Reid, Martin Hirst, Steven Jones, Jennifer Chan, Donna Senger, Jason Berman, Seth J. Parker, Jonathan Bush, Caron Strahlendorf, Rebecca Deyell, CHINTEN Lim, and Philipp Lange

Corresponding author(s): Philipp Lange (philipp.lange@ubc.ca) , CHINTEN Lim (cjlim@mail.ubc.ca), Rebecca Deyell (rdeyell@cw.bc.ca)

Review Timeline:

Submission Date:	27th Aug 24
Editorial Decision:	30th Sep 24
Revision Received:	30th Jan 25
Editorial Decision:	17th Feb 25
Revision Received:	25th Feb 25
Accepted:	26th Feb 25

Editor: Zeljko Durdevic

Transaction Report:

30th Sep 2024

Dear Dr. Lange,

Thank you for the submission of your revised manuscript to EMBO Molecular Medicine. We have now received feedback from the three reviewers who agreed to evaluate your manuscript. As you will see from the reports below, all three referees acknowledge interest of the study but also raise serious and largely overlapping concerns particularly regarding the part about incorporation of proteomics into precision medicine.

Taking referee concerns in consideration it is clear that publication of the paper cannot be considered at this stage. After our cross-commenting session referees agreed that a shorter report focusing on the case report, toning down the part about the implementation of the approach in precision oncology and clearly stating the limitations of the study in that regard would be suitable for further consideration in EMBO Molecular Medicine. Referees also agreed that all technical concerns should be addressed in a major revision. If you decide to follow this path, I would like you to reformat and rewrite your manuscript as a scientific report (3 figures, ~22000 characters), for more information please check our "Author Guidelines".

<https://www.embopress.org/page/journal/17574684/authorguide#reportsarticleguide>. If you would like to discuss further the points raised by the referees, I am available to do so via email or video. Let me know if you are interested in this option.

We would welcome the submission of a revised version within three months for further consideration. Please let us know if you require longer to complete the revision.

I look forward to receiving your revised manuscript.

Yours sincerely,

Zeljko Durdevic

We require:

- 1) A .docx formatted version of the manuscript text (including legends for main figures, EV figures and tables). Please make sure that the changes are highlighted to be clearly visible.
- 2) Individual production quality figure files as .eps, .tif, .jpg (one file per figure). For guidance, download the 'Figure Guide PDF': (<https://www.embopress.org/page/journal/17574684/authorguide#figureformat>).
- 3) A .docx formatted letter INCLUDING the reviewers' reports and your detailed point-by-point responses to their comments. As part of the EMBO Press transparent editorial process, the point-by-point response is part of the Review Process File (RPF), which will be published alongside your paper.

- 4) A complete author checklist, which you can download from our author guidelines (<https://www.embopress.org/page/journal/17574684/authorguide#submissionofrevisions>). Please insert information in the checklist that is also reflected in the manuscript. The completed author checklist will also be part of the RPF.
- 5) Please note that all corresponding authors are required to supply an ORCID ID for their name upon submission of a revised manuscript.
- 6) It is mandatory to include a 'Data Availability' section after the Materials and Methods. Before submitting your revision, primary datasets produced in this study need to be deposited in an appropriate public database, and the accession numbers and database listed under 'Data Availability'. Please remember to provide a reviewer password if the datasets are not yet public (see <https://www.embopress.org/page/journal/17574684/authorguide#dataavailability>).
- In case you have no data that requires deposition in a public database, please state so in this section. Note that the Data Availability Section is restricted to new primary data that are part of this study.
- 7) For data quantification: please specify the name of the statistical test used to generate error bars and P values, the number (n) of independent experiments (specify technical or biological replicates) underlying each data point and the test used to calculate p-values in each figure legend. The figure legends should contain a basic description of n, P and the test applied. Graphs must include a description of the bars and the error bars (s.d., s.e.m.). See also 'Figure Legend' guidelines: <https://www.embopress.org/page/journal/17574684/authorguide#figureformat>
- 8) At EMBO Press we ask authors to provide source data for the main manuscript figures. Our source data coordinator will contact you to discuss which figure panels we would need source data for and will also provide you with helpful tips on how to upload and organize the files.
- 9) Our journal encourages inclusion of *data citations in the reference list* to directly cite datasets that were re-used and obtained from public databases. Data citations in the article text are distinct from normal bibliographical citations and should directly link to the database records from which the data can be accessed. In the main text, data citations are formatted as follows: "Data ref: Smith et al, 2001" or "Data ref: NCBI Sequence Read Archive PRJNA342805, 2017". In the Reference list, data citations must be labeled with "[DATASET]". A data reference must provide the database name, accession number/identifiers and a resolvable link to the landing page from which the data can be accessed at the end of the reference. Further instructions are available at .
- 10) We replaced Supplementary Information with Expanded View (EV) Figures and Tables that are collapsible/expandable online. A maximum of 5 EV Figures can be typeset. EV Figures should be cited as 'Figure EV1, Figure EV2' etc... in the text and their respective legends should be included in the main text after the legends of regular figures.
- For the figures that you do NOT wish to display as Expanded View figures, they should be bundled together with their legends in a single PDF file called *Appendix*, which should start with a short Table of Content. Appendix figures should be referred to in the main text as: "Appendix Figure S1, Appendix Figure S2" etc.
 - Additional Tables/Datasets should be labeled and referred to as Table EV1, Dataset EV1, etc. Legends have to be provided in a separate tab in case of .xls files. Alternatively, the legend can be supplied as a separate text file (README) and zipped together with the Table/Dataset file.

12) Author contributions: You will be asked to provide CRediT (Contributor Role Taxonomy) terms in the submission system.

These replace a narrative author contribution section in the manuscript.

13) A Conflict of Interest statement should be provided in the main text.

14) Every published paper now includes a 'Synopsis' to further enhance discoverability. Synopses are displayed on the journal webpage and are freely accessible to all readers. They include a short stand first (maximum of 300 characters, including space) as well as 2-5 one-sentences bullet points that summarizes the paper. Please write the bullet points to summarize the key NEW findings. They should be designed to be complementary to the abstract - i.e. not repeat the same text. We encourage inclusion of key acronyms and quantitative information (maximum of 30 words / bullet point). Please use the passive voice. Please attach these in a separate file or send them by email, we will incorporate them accordingly.

15) Include a Reagents and Tools Table as part of the Methods section, which can be downloaded from our author guidelines (<https://www.embopress.org/page/journal/17574684/authorguide#structuredmethods>)

***** Reviewer's comments *****

Referee #1 (Comments on Novelty/Model System for Author):

The fundamental conundrum this manuscript presents is as follows. The work presented in the paper is, technically, of a high standard and it is difficult to provide additional suggest regarding technical aspects beyond those I have outlined in my comments to the authors. The real issue is that whilst going to almost heroic lengths in the modelling, this is still a report of the potential feasibility of proteomics in precision medicine (of which this would be one of the first reports) through the lens of an individual case-report, in which there was an underwhelming clinical response signal. Further, the nature of the controls and comparators required for proteomics to be a future methodology applied in precision medicine are challenging, and in this study, mostly lacking.

Thus I believe there are some clear merits to the work, the novelty and innovation presented are worthy, but the medical impact is low and unproven. One might justify publication if it were the novelty and innovation that were being emphasised (which is not the case in this report), but that would be balanced against the short-comings described. My judgement is that it is not ready for publication, but I am am willing to review a revised version in the other reviewers and editorial opinions are more favourable than mine.

Referee #1 (Remarks for Author):

This manuscript from Barnabas and colleagues presents what seems to be a feasibility study of the incorporation of proteomics into precision medicine in pediatric cancer, through the lens of a case report. The authors describe the proteomics that defined a potential pathway dependency in a rare tumour, when to considerable lengths to demonstrate the potential of targeting 1C metabolism (sertraline dependence) in PDX models of this patient, and then report on the clinical response. There are many features to admire about the work, including the thoroughness of the biology presented and the modelling. Further, this is also one of the first "case-reports" describing the incorporation of proteomics into precision medicine and suggests, on the basis of this case, that it is feasible. Yet this also gets to some of the difficulties with the manuscript. Is this a demonstration of feasibility of an arm of omics to be added to sequencing and in vitro screening, or is this a one-off report? The former is far more impactful but would obviously require more than a single case, since there are important questions about repeatability and controls to be addressed. If the later, then the case report whilst good, is much less compelling. This is more so since the clinical response was marginal (the FLT3 inhibitor seemed to have a more favourable response until resistance, and this was selected by mutation detection). Further, there are no controls for positive and negative response to the selected drug - in other words, how is the specificity of the link between the elevated 1C metabolism peptides and true drug responses in cancer established?

Some additional comments are provided below

- One of the messages that seems to be emerging from precision medicine studies is that the response to targeted therapies is best when there exists a strong mechanistic link between the genomic (or perhaps proteomic) aberration and the targeted drug. Here, an unsubstantiated (although intriguing) link between an outlier peptide/pathway expression identified using proteomics us postulated to define a dependency on sertraline, and a potential therapy defined in this way. What other controls could be added (beyond just this sample and the PDX models which all derive from the same tissue) to test this hypothesis more convincingly?
- Where there any features in the proteomic data that may be associated with the acquired resistance to the FLT3 inhibitor?
- With respect to upregulation of SHMT2, this is compared to normal tissue extracted from the same sections. This is a strong

internal control. However, there are no other external controls to understand the range of expression of the peptide in tumours, how repeatably detectable it is. Such controls are, one appreciates, challenging, but critical to interpretation.

Referee #2 (Comments on Novelty/Model System for Author):

The novelty of the part on the single SETTLE patient is high. The Novelty regarding application of proteomics in a precision oncology program is medium. The same applies to Medical impact.

I feel that the authors try to sell several stories in this manuscript. The first is a case report on a single SETTLE patient - a follow-up of a study that they had published in *Fetal Pediatr Pathol* 38, 399-405 (2019). The second is about their approach of applying proteomic and functional genomic technologies for potential integration in their clinical precision oncology program. They might better stick to one story, and I believe that their case report would be the more appropriate one. However, this might not be suited so well for *EMBO Mol Med*.

Referee #2 (Remarks for Author):

Barnabas and colleagues present a case report study, where they comprehensively analyzed and describe a single patient suffering from an advanced spindle epithelial tumor with thymus-like elements (SETTLE). This patient was enrolled in the Canadian PROFYLE clinical study aiming at molecular analysis of tumors to provide more efficient treatment options of children, adolescents, and young adults. The same patient had been molecularly analyzed before (without proteomics) and respective findings were published as a case report some years back (reference 25).

On top of routine sequencing of nucleic acids (genetic/genomic alterations, RNA-expression), the authors here applied mass spectrometry to assess the abundance of proteins and exploit the resulting data towards identifying additional therapeutic targets. This prioritized SHMT2 and 1C carbon metabolism as such potential target, which was then initially verified in rodent PDX, CAM, and zebrafish models. The authors continued with stable isotope labeling and metabolomics measurements. Finally, the patient received an inhibitor of SHMT2 and showed reduced tumor growth, however, no response according to RECIST criteria.

This is a very deep analysis of a single case presenting with a pediatric SETTLE tumor. The authors claim that their study shows that proteome-guided and functional precision oncology are feasible and valuable complements to the current genome-driven precision oncology practices. The application of proteomics in precision oncology has been described before, however, indeed mostly in retrospective studies (several references in the current study). While the analysis of the single SETTLE tumor is indeed comprehensive, even though I miss an integrative analysis of genomic and proteomic features, this study is likely impossible to extend to a larger patient cohort. Establishment of PDX models is time consuming and other approaches, like organoids seem to be more promising, also towards screening of larger drug libraries. The metabolomics part could become a module to be applied in cases where metabolic enzymes might be drivers of respective disease and putative therapeutic targets.

Other:

In the abstract, they write that they identified elevated protein level of SHMT2 - compared to what baseline level? It appears that the authors compared expression levels in tumors and metastases to normal surrounding (e.g., lung) tissue. In other precision oncology studies, expression levels within individual tumors are compared with expression levels in tumors within a larger cohort, to establish ranked lists and grouping some individual's expression state relative to the other tumors. Thereby, recommendations for particular targeted drugs can be based on previous successful application of the same drugs in other patients with similar or even lower expression levels. The list of expression states in such a ranked list increases as the number of patients in the cohort increases. In the current SETTLE patient it is not described well enough what cell types were present and then compared in the tumor and normal specimen, respectively (the authors mention macrodissection).

The authors of the current study seem to tell two stories in one paper: 1. they present a very detailed description of a single case with a SETTLE tumor and 2. they try to convince the reader that their approach is suited to be implemented in a clinical precision oncology program. The first story is indeed convincing even though the clinical benefit for the patient seems to be limited - which could be expected given the advanced state of the disease. The second story would require a more comparative investigation with other functional genomic applications. The time needed for establishment of a PDX model (the authors speak of 6-8 months, paragraph "Mapping tumor histology...") is way too long for implementation in a clinical program. The authors might consider establishing and testing organoid models instead of PDX (e.g., PMID: 39019014, 39229726, 38593780). Is the described cryopreservation of primary tumor tissues associated with the long time it takes for tumor engraftment? In the abstract, the authors write that their analysis was completed within 2 months and [thus] ahead of a molecular tumor board. The timeline from biopsy to discussion of findings in a tumor board appears a little long, particularly, when patients have advanced and progressive disease.

The authors state that they received FFPE sections from pathology. Similar to nucleic acid analysis, also proteomic analysis should benefit from analyzing fresh tissue sections instead of FFPE tissues. It is not clear from the text what parameters were used during macro-dissection (isolation of just epithelial cells or larger tumor areas?). It is thus not clear what kinds of cell types

were analyzed from tumor and from normal tissues.

The authors speak of the abundance of 4703 proteins in metastatic nodules and adjacent normal lung regions. Later, they write about 6921 proteins that are claimed to demonstrate in-depth proteome coverage (Fig. 1F). The dynamic range achieved in this analysis is indeed impressive, however, the number of expressed proteins could be expected higher (in the order of 10^5 - $>11,000$), depending on the cell types/tumor cell content having been sampled. Am I correct to assume that about half of the cancer-associated proteins the authors had previously assembled ($n=349$) were not quantified in the current study ($n=154$). Are the missing proteins not expressed in the tissue/cell types analyzed or are they not detected? Datasets, like CPTAC and their own RNA-seq data might provide clues.

The authors determined IC50 values for SETTLE cells and found these to be between 5.5 and 10.1 μM (Fig. 4C). How do these IC50 values compare to drug effects in normal (lung) tissue/mice? What are potential side effects of the drug at this concentration? 'Sertraline HCl (32 $\mu\text{mol/kg}$ i.p.) inhibits serotonin uptake into striatal synaptosomes from rats by more than 50%' Selleckchem (<https://www.selleckchem.com/datasheet/sertraline-hcl-S405302-DataSheet.html>). The authors should compare their experimentally identified IC50 values with some reference.

The authors use the heading: "Sertraline and trimethoprim combination as a potential therapeutic approach for SETTLE tumors". The authors analyzed one patient. Can they indeed generalize for SETTLE as an entity? Their findings and rationale may be very relevant for other patients and entities that rely on 1C carbon metabolism - the authors cite (ref. 37) a study having tested serine-dependent cell models of breast cancer, and present drug-synergy data having been obtained using two TNBC cell lines (this could become kind of a third story).

In Fig. 6D the decline of monthly growth rates is shown. Fig. 6D gives percentages/month with mostly steep declines upon setraline therapy (pink) compare to pre setraline (black), and further decreases post setraline (yellow). How were monthly tumor growth rates assessed prior to setraline treatment?

Minor:

In the introduction, the authors might cite some study that has described the outcome specifically for pediatric patients, e.g., PMID:34492587. It should be noted that those improvements have mostly not been achieved based on genomic profiling though. The current challenge in pediatric oncology is relapsed and metastatic disease as those tumors are mostly refractory to therapy.

The authors state that INFORM has not reported "on the response or survival status [of patients] (13)." The authors might consider PMID:37364231

Still in the introduction, the authors write about "challenges arising from the limited quantity and of available viably cryopreserved tissues ... (15)." The authors should be aware that their reference 15 only lists studies that used fresh tissues for generation of PDX models. Cryopreserved tissue has rarely been reported for generation of PDX models (e.g., PMID: 33910584). Utilization of fresh tissues might speed up engraftment of PDXs.

Figure 1D lists 29 proteins, not 14. It should be explained that the remaining proteins did not map to respective interventional baskets.

The authors found significant increases in PTEN levels. Increased abundance of PTEN might be associated with enhanced growth factor PI3K-mTOR signaling activity in the tumor compared to adjacent normal lung tissue. An assessment of pathway activities (e.g., using RNA-seq data) might be warranted.

The authors write that they were able to rapidly generate CAM-PDXs [which] afforded an opportunity to rapidly evaluate targeting the 1C metabolic pathway. How rapid was generation and testing of this CAM-PDX model?

Referee #3 (Comments on Novelty/Model System for Author):

The results with the CAM xenograft model were difficult to interpret from the figures shown and the response to the proposed targeted therapy was modest. Since this was an 'n of one' study, it is impossible to tell if the model was insufficiently sensitive, or if the tumor was heterogeneous and difficult to model. These points are in my Comments to Authors

Referee #3 (Remarks for Author):

The manuscript 'Proteomics and personalized patient-derived xenograft models identify treatment opportunities for a progressive malignancy within a clinically actionable timeframe and change care' by Barnabas et al is a very interesting and potentially consequential demonstration of two important points: 1) that adding measurements of proteins (proteomics) to traditional 'precision oncology' measurements of DNA and RNA can contribute significant new insights to individualized therapeutic interventions, and 2) that adding proteomic measurements is clinically feasible within a two week time frame. Additionally, they propose a workflow for establishing a patient-derived xenograft in a chick allantoic membrane (CAM) model system that can provide ex vivo results to a Molecular Tumor Board within a two month time period.

While the potential power of adding proteomics and ex vivo testing on patient derived xenografts in the context of precision oncology is truly exciting and highly transformative, the current work has several significant limitations. This is an 'n of one' study, with a single patient, so no comparison can be drawn between this approach and the more traditional nucleic acids only approaches. Because it is an n of one study and patient-derived materials are seriously limited, insufficient evidence is

presented to truly validate the CAM xenograft system as an adequate model system for identifying individualized therapeutic strategies. The experimental evidence provided to support ex vivo testing is weak, and it is impossible to tell whether that is because the model system has limited efficacy or because the n of one patient simply had a highly heterogeneous and treatment resistant tumor by the time that proteomics was introduced, at the third recurrence as lung metastases.

The authors are well aware of the limitations imposed by the 'n of one' design, but they justify submitting this preliminary result in order to stimulate interest in the approach and encourage other investigators with access to appropriate patients and clinical trials to attempt the same approach. This is a highly laudable motive, particularly as that type of multi-center, multi-investigator, multi-disease data is precisely what will be needed to either confirm or reject the clinical utility of adding proteomics to target identification and CAM xenografts to target validation in a practical clinical context. That said, there are some changes that could be made to the current manuscript to improve its usefulness to other researchers in the field and help avoid enthusiasm beyond what has actually been demonstrated.

Suggested Improvements:

1. In reading the text linearly, there were several questions critical to evaluation of the results that were not addressed in the text, but were handled nicely in the Figures. The text should either address these questions directly or clearly direct the reader to the appropriate figure for the following:

a. The precise timeline of initial nucleic acid testing, administration of nucleic acid-based targeted therapies, the various relapses, and the addition of proteomic measurements over the course of the patient's disease is unclear in the text. All of that is handled very nicely in Figure 1B. However, the figure legend for 1B should be explicit that the proteomic analysis of PR and R1 was retrospective.

b. Similarly, Figure 1A does a great job of delineating the time required for each step compared to the standard clinical evaluation process; this could have been clarified in the text.

2. When describing the precision oncology analysis of the R1 tumor, please specify that the 'increased expression' observed for specific genes was at the RNA level. Proteins are gene products, too, and not all readers will assume 'gene expression' is automatically mRNA.

3. Since the investigators did a retrospective proteomic analysis of tumor from R1, which also has genomics and transcriptomics, a table comparing the changes observed in the R1 tumor at the DNA, RNA, and protein levels would have been informative in evaluating what is added by proteomics. This could be done by adding proteins to Supplemental Table 1 or by constructing a new table that highlighted only the changes.

4. The quality of the proteomic data shown is good, but given the only 44% overlap between the DIA results and known 'pediatric cancer proteins' in Fig 1F, information on the subcellular distribution of the identified proteins would have been helpful, specifically, whether there was enrichment of cytoplasmic versus membrane proteins. Suppl Figure 1C shows biological processes, but not subcellular localization.

5. Were any PTMs identified in the proteomic data, particularly phosphopeptides? Given the somewhat lower than expected representation of kinases in the observed known proteins, this could be relevant to the ability to identify potential targets.

6. It is really difficult for the reader to interpret the micrographs in Figure 4A. The reflections off the xenograft and the differences in lighting are truly distracting, and it is not even clear what is the tumor in the D1 sertraline image. In comparison, the micrographs in Suppl Figure 5A are much easier to interpret. Arrows should be used to define the margins of the xenograft. Better images would help. I had to look up a review article on CAM xenografts to interpret what I was seeing.

7. One is left with the overall impression that the utility of the CAM xenograft model is somewhat overstated in the manuscript. The CAM xenograft histology in Figure 3C is very different from both the patient's R2 (Figure 3A) and NSG-PDX (Figure 3B); interestingly, the image in Suppl Fig 3A is much more convincing. The effects of sertraline on tumor size in the CAM-PDX model are not highly impressive, and don't show statistical significance until Day 5. There is no synergy between sertraline and trimethoprim; the results aren't even additive - not much of an incentive to try combination therapies. A more honest assessment of the preliminary nature of the ex vivo drug testing in the CAM-PDX model with some suggestions for improvement would have bolstered confidence in the potential clinical utility of this workflow. Given the modest effects of sertraline in the CAM-PDX data shown, it is not at all surprising that the patient only showed a reduction in tumor growth rate in response to sertraline treatment. Had there been an example of a more impressive ex vivo response correlated with a more significant clinical outcome, it actually would have enhanced confidence in the ability of the CAM-PDX system to recapitulate the patient's tumor biology.

Response to reviews.

Referee #1 (Comments on Novelty/Model System for Author):

The fundamental conundrum this manuscript presents is as follows. The work presented in the paper is, technically, of a high standard and it is difficult to provide additional suggest regarding technical aspects beyond those I have outlined in my comments to the authors. The real issue is that whilst going to almost heroic lengths in the modelling, this is still a report of the potential feasibility of proteomics in precision medicine (of which this would be one of the first reports) through the lens of an individual case-report, in which there was an underwhelming clinical response signal. Further, the nature of the controls and comparators required for proteomics to be a future methodology applied in precision medicine are challenging, and in this study, mostly lacking.

Thus I believe there are some clear merits to the work, the novelty and innovation presented are worthy, but the medical impact is low and unproven. One might justify publication if it were the novelty and innovation that were being emphasised (which is not the case in this report), but that would be balanced against the short-comings described. My judgement is that it is not ready for publication, but I am willing to review a revised version in the other reviewers and editorial opinions are more favourable than mine.

We appreciate the reviewer's careful evaluation of the manuscript and noting that it is of high standard and one of the first reports of proteomics in precision medicine. We agree with the frank assessment that it is limited to a single case with limited clinical response and that key controls and comparators remain challenging. To overcome the challenge of missing proteomic comparators we had conducted orthogonal validation of the target overexpression by immunohistochemistry on tumor microarrays. This effectively provides the same ranked assessment typically used in sequencing-based precision medicine target prioritization. To address the limitation of a single case we have now more rewritten the manuscript as a case-report.

Referee #1 (Remarks for Author):

This manuscript from Barnabas and colleagues presents what seems to be a feasibility study of the incorporation of proteomics into precision medicine in pediatric cancer, through the lens of a case report. The authors describe the proteomics that defined a potential pathway dependency in a rare tumour, when to considerable lengths to demonstrate the potential of targeting 1C metabolism (sertraline dependence) in PDX models of this patient, and then report on the clinical response. There are many features to admire about the work, including the thoroughness of the biology presented and the modelling. Further, this is also one of the first "case-reports" describing the incorporation of proteomics into precision medicine and suggests, on the basis of this case, that it is feasible. Yet this also gets to some of the difficulties with the manuscript. Is this a demonstration of feasibility of an arm of omics to be added to sequencing and *in vitro* screening, or is this a one-off report? The former is far more impactful but would obviously require more than a single case, since there are important questions about repeatability and controls to be addressed. If the later, then the case report whilst good, is much less compelling. This is more so since the clinical response was marginal (the FLT3 inhibitor seemed to have a more favourable response until resistance, and this was selected by mutation detection). Further, there are no controls for positive and negative response to the selected drug - in other words, how is the specificity of the link between the elevated 1C metabolism peptides and true drug responses in cancer established?

We appreciate the careful assessment by the reviewer. Given the limited availability of viable patient-derived SETTLE tumor, we had limited drug testing of the SETTLE tumor to sertraline (Fig. 3A,B) and SHIN1 (Fig. 3C) as CAM PDXs, which exhausted all available primary tumor material at the time. In parallel, the tumor achieved reasonable expansion as PDXs in immune deficient NSG mice, affording additional tumor for subsequent follow up drug assays in CAM and zebrafish xenografts (Fig. EV3D,E), as well as *in vitro* in short term culture assays (Fig. 3D,E and Fig. EV3C,F). To address the concern of demonstrating a positive or negative response using PDX model, we have now included sertraline drug testing of CAM xenografts of MDA-MB-468 and MDA-MB-231, respectively, breast tumor cell lines previously demonstrated to be serine-addicted and non-addicted (PMID:33203732). The newly added data (Fig. EV3B) provides supporting evidence that the serine-addicted MDA-MB-468 tumors exhibited enhanced sensitivity (or positive response) to sertraline. To confirm the specificity of the link between the elevated 1C metabolism and sertraline treatment, we had conducted stable isotope tracing experiments that clearly showed the on-target effect of the treatment (Fig.3 D,E). We have now revised the text to communicate this more clearly.

Some additional comments are provided below

- One of the messages that seems to be emerging from precision medicine studies is that the response to targeted therapies is best when there exists a strong mechanistic link between the genomic (or perhaps proteomic) aberration and the targeted drug. Here, an unsubstantiated (although intriguing) link between an outlier peptide/pathway

expression identified using proteomics was postulated to define a dependency on sertraline, and a potential therapy defined in this way. What other controls could be added (beyond just this sample and the PDX models which all derive from the same tissue) to test this hypothesis more convincingly?

Pre-clinical reports in breast and lung cancer provided led us to hypothesize that there was in fact a strong mechanistic link between the SHMT2 elevation, cancer growth and survival and sertraline treatment. To confirm that this mechanistic link does exist in the patient's malignant cells we conducted stable isotope serine and glucose tracing experiments with and without sertraline treatment (Fig.3 D,E). We have re-written this section to communicate this more effectively. To further strengthen this mechanistic link we now also include similar experiments on cell lines with and without serine addition (new Fig. EV3B).

- Where there any features in the proteomic data that may be associated with the acquired resistance to the FLT3 inhibitor?

This is an excellent question. We had initially decided not to discuss this in the manuscript to keep it focussed on the latest biopsy. In our process of re-writing the manuscript as a case-report as requested we are now providing additional extensive analyses. At both timepoints FLT3, while mutated, has low RNA expression levels and was below detection or absent at the protein level. Sorafenib is a broad specificity tyrosine kinase inhibitor and it is therefore likely that it will have acted through inhibition of one or several of the RTKs that had been found to be elevated at R1 but showed reduced RNA expression at R3. This may in part explain the acquired resistance to FLT3 as we now discuss in the revised manuscript. This data has been added as new Appendix Figure S2A,B

- With respect to upregulation of SHMT2, this is compared to normal tissue extracted from the same sections. This is a strong internal control. However, there are no other external controls to understand the range of expression of the peptide in tumours, how repeatably detectable it is. Such controls are, one appreciates, challenging, but critical to interpretation.

The reviewer is correct that the primary mass spectrometry-based quantification of the SHMT2 increase was limited to normal tissue from the same sections. We then confirmed that SHMT2 was consistently increased at earlier disease timepoints of the same patient. This included additional normal tissue controls so that we could provide mass spectrometry-based evidence for elevation of SHMT2 in 4 different biopsies compared to adjacent normal lung and thyroid tissue.

We also found SHMT2 to be consistently elevated in 4 additional SETTLE patients as shown here. As we do not have viable cells from these additional cases we chose not to include this information in the manuscript to keep it focussed on the individual case as requested by the reviewers and editor.

Figure for referees not shown.

We agree with the reviewer that these controls are critical to the interpretation. In addition to these strong internal controls, we had performed extensive orthogonal validation both within the patient and across other tumors. SHMT2 was measured by immunohistochemistry across 15 tumor types from 80 cases (Fig. 2D). This clearly confirmed the elevation within the patient compared to adjacent normal and importantly also provided the controls pointed out by the reviewer. They clearly showed that SHMT2 abundance ranked high in the patient compared to most other tumor types. Importantly it was also higher than in breast and lung tumors for which pre-clinical evidence for sertraline sensitivity had previously been established.

Together we argue that these data provide strong evidence for elevation of SHMT2, both compared to patient normal lung and thyroid tissue as well as compared to other tumor tissues that show sertraline sensitivity.

We have now improved the presentation of the figures and re-wrote the respective section to provide a more accessible account of the conducted controls and rigour of the target classification.

Referee #2 (Comments on Novelty/Model System for Author):

The novelty of the part on the single SETTLE patient is high. The Novelty regarding application of proteomics in a precision oncology program is medium. The same applies to Medical impact.

I feel that the authors try to sell several stories in this manuscript. The first is a case report on a single SETTLE patient - a follow-up of a study that they had published in *Fetal Pediatr Pathol* 38, 399-405 (2019). The second is about their approach of applying proteomic and functional genomic technologies for potential integration in their clinical precision oncology program. They might better stick to one story, and I believe that their case report would be the more appropriate one. However, this might not be suited so well for *EMBO Mol Med*.

Referee #2 (Remarks for Author):

Barnabas and colleagues present a case report study, where they comprehensively analyzed and describe a single patient suffering from an advanced spindle epithelial tumor with thymus-like elements (SETTLE). This patient was enrolled in the Canadian PROFYLE clinical study aiming at molecular analysis of tumors to provide more efficient treatment options of children, adolescents, and young adults. The same patient had been molecularly analyzed before (without proteomics) and respective findings were published as a case report some years back (reference 25).

On top of routine sequencing of nucleic acids (genetic/genomic alterations, RNA-expression), the authors here applied mass spectrometry to assess the abundance of proteins and exploit the resulting data towards identifying additional therapeutic targets. This prioritized SHMT2 and 1C carbon metabolism as such potential target, which was then initially verified in rodent PDX, CAM, and zebrafish models. The authors continued with stable isotope labeling and metabolomics measurements. Finally, the patient received an inhibitor of SHMT2 and showed reduced tumor growth, however, no response according to RECIST criteria.

This is a very deep analysis of a single case presenting with a pediatric SETTLE tumor. The authors claim that their study shows that proteome-guided and functional precision oncology are feasible and valuable complements to the current genome-driven precision oncology practices. The application of proteomics in precision oncology has been described before, however, indeed mostly in retrospective studies (several references in the current study). While the analysis of the single SETTLE tumor is indeed comprehensive, even though I miss an integrative analysis of genomic and proteomic features, this study is likely impossible to extend to a larger patient cohort. Establishment of PDX models is time consuming and other approaches, like organoids seem to be more promising, also towards screening of larger drug libraries. The metabolomics part could become a module to be applied in cases where metabolic enzymes might be drivers of respective disease and putative therapeutic targets.

We thank the reviewer for their careful assessment of the manuscript. As suggested we have now focussed on the case report and included additional integrative analysis of genomic and proteomic features as detailed below.

Other:

In the abstract, they write that they identified elevated protein level of SHMT2 - compared to what baseline level? It appears that the authors compared expression levels in tumors and metastases to normal surrounding (e.g., lung) tissue. In other precision oncology studies, expression levels within individual tumors are compared with expression levels in tumors within a larger cohort, to establish ranked lists and grouping some individual's expression state relative to the other tumors. Thereby, recommendations for particular targeted drugs can be based on previous successful application of the same drugs in other patients with similar or even lower expression levels. The list of expression states in such a ranked list increases as the number of patients in the cohort increases.

We thank the reviewer for alerting us to the fact that this was not well described in our initial manuscript. As detailed in the response to reviewer 1 we had indeed followed this well-established concept. After identifying SHMT2 elevation in the patient's primary tumor and all recurrent/relapse biopsies compared to the internal baseline in adjacent normal thyroid and lung tissue, we ranked its abundance in 15 tumor types from 80 cases using IHC (Fig. 2D). This showed clear outlier expression in the patients tumor well above the levels in breast and lung tumors for which pre-clinical animal studies demonstrated sertraline efficacy. We have now rewritten these sections to make this more accessible to the reader.

In the current SETTLE patient it is not described well enough what cell types were present and then compared in the tumor and normal specimen, respectively (the authors mention macrodissection).

We fully agree that this lacks clarity. To provide a detailed account of the regions and cell types that were selected for dissection we have now expanded the respective section in the Materials and Methods it now reads:

"After deparaffinization and H&E staining, the slides were macrodissected into the tumor (including spindle and epithelial elements) and adjacent normal tissue by a pathologist. The adjacent normal tissue for metastatic lesions captured lung parenchyma including predominantly alveolar epithelium, vascular structures, and a small amount of pleural mesothelium and associated fibroblasts. Small areas of respiratory bronchiole epithelium, smooth muscle, and lymphoid tissue were also included. Red blood cells were present, as were focal areas of alveolar macrophages. The normal tissues were not selected to have minimal non-epithelium elements samples. The adjacent normal for the primary tumor encompassed thyroid tissue including thyroid follicles, stromal fibroblasts and vascular structures but excluding adjacent skeletal muscle."

The authors of the current study seem to tell two stories in one paper: 1. they present a very detailed description of a single case with a SETTLE tumor and 2. they try to convince the reader that their approach is suited to be implemented in a clinical precision oncology program. The first story is indeed convincing even though the clinical benefit for the patient seems to be limited - which could be expected given the advanced state of the disease. The second story would require a more comparative investigation with other functional genomic applications. The time needed for establishment of a PDX model (the authors speak of 6-8 months, paragraph "Mapping tumor histology...") is way too long for implementation in a clinical program.

We appreciate this comment and the opportunity to better outline the pre-clinical phases of this study that resulted in clinical action within a reasonable time frame. As outlined in the schematic shown in Fig. 1A-B, we used R2 tumor material to successfully establish PDXs in both the chick CAM and NSG mouse models. While these were performed in parallel, the time frames involved to achieve engraftment and propagation were significantly different. For CAM PDXs, we completed between 4-5 cycles of CAM engraftment and propagation, with the last cycle dedicated to drugging assays (this data shown in Fig. 3A-C); all occurring within 6 weeks following thawing of the patient derived cryopreserved material. In contrast, the NSG PDXs required 6-8 months to establish a palpable tumor, which at sacrifice resulted in significant expansion, and replenishment, of the now already exhausted starting material. In this manner, the NSG PDXs afforded additional material for further pre-clinical assays, including follow up drug assays in CAM and zebrafish PDXs (Fig. EV3D,E), as well as *in vitro* in short term culture assays (isotope tracing in Fig. 3D,E; and drug assays Fig. EV3C,F). We have now improved the text to communicate the differences between the complimentary PDX models used more clearly; including the following:

"We used viably cryopreserved R2 tissue to establish and evaluate drug response as CAM PDXs. In parallel, R2 tumor was also established as mouse PDXs, affording expansion and archiving of SETTLE cells for additional CAM and zebrafish ex vivo drug studies, and to confirm on-target activity by isotope tracing in vitro."

The authors might consider establishing and testing organoid models instead of PDX (e.g., PMID: 39019014, 39229726, 38593780). Is the described cryopreservation of primary tumor tissues associated with the long time it takes for tumor engraftment?

We thank the reviewer for the suggestion of establishing patient-derived tumor organoids (PDO), which, if successful, would have provided sufficient replenishable material for many pre-clinical studies and within a rapid timeframe. As you know, the ability to establish PDO is enabled by defined organoid culture protocols, each of which is specialized for the tumor/tissue type in question (media, growth factors and so on). Outside of certain brain and kidney tumors, and some forms of sarcoma, there have been few defined PDO protocols described for pediatric tumors, and we would have little basis to establish PDO from this very rare SETTLE tumor. We have added text to the Conclusion /Discussion section that briefly compared the utility of the different *ex vivo* (PDX, PDO) models.

Our choice to utilize the CAM PDX model was predicated on prior successes with a range of pediatric solid tumors. The amount of viable tissue that remains available following the required clinical pathology testing is often limited for pediatric tumors limiting the feasibility of testing multiple culture conditions and rendering the CAM PDX model as an attractive first option. As mentioned earlier, the viably cryopreserved R2 SETTLE tumor required only one cycle of xenografting in CAM (5-7 days post implantation) to observe growth of the microtumors, subsequently confirmed by IHC and H&E analysis (Appendix Supplementary Information). Starting with an aliquot of the same R2 tumor, the successful generation of PDXs in NSG mice required the longer 6-8 months time frame. While we have no basis for comparison, it has been suggested that engraftment efficiency (in mice) is not better or worse when comparing cryopreserved and fresh tumors (PMID:33910584). Incidentally, we attempted but failed to generate CAM-PDXs from fresh R3 SETTLE tumor.

In the abstract, the authors write that their analysis was completed within 2 months and [thus] ahead of a molecular tumor board. The timeline from biopsy to discussion of findings in a tumor board appears a little long, particularly, when patients have advanced and progressive disease.

The proteomics report with IHC validation was completed and submitted within 2 weeks. Since this was a slowly progressing tumor, the genome analysis and immediate discussion at an MTB was of lower urgency and the MTB was held two months after biopsy. We agree that the two month timespan to MTB seen here would be problematic for many rapidly progressing tumors and is not the norm in the PROFYLE study. Encouragingly, the 2 weeks we required for the proteomics analysis and validation is sufficiently fast to provide timely information for faster progressing cases. The MTB discussion included the *ex vivo* sertraline treatment data already completed using the chick CAM PDX model. We have now stated this more carefully and clearly.

The authors state that they received FFPE sections from pathology. Similar to nucleic acid analysis, also proteomic analysis should benefit from analyzing fresh tissue sections instead of FFPE tissues. It is not clear from the text what parameters were used during macro-dissection (isolation of just epithelial cells or larger tumor areas?). It is thus not clear what kinds of cell types were analyzed from tumor and from normal tissues.

As described above we have now added details about the dissected tumor areas and cell types to the methods section. To clarify the use of FFPE rather than fresh biopsies we have expanded the supplementary text now stating: *"It is now well established that proteomics on FFPE tissues faithfully recapitulates the proteome with similar coverage and good correlation to fresh frozen tissue sections (PMID: 30087585), and we previously confirmed that our ASAP FFPE proteome processing allows for unbiased proteome profiling (PMID 36724070). In contrast to fresh sectioning, FFPE biopsy processing is a routine element of the clinical workflow and robust to delays, interruptions, and transfer between institutions, making it the preferred sample source."*

The authors speak of the abundance of 4703 proteins in metastatic nodules and adjacent normal lung regions. Later, they write about 6921 proteins that are claimed to demonstrate in-depth proteome coverage (Fig. 1F). The dynamic range achieved in this analysis is indeed impressive, however, the number of expressed proteins could be expected higher (in the order of 10^5 - $>10^6$), depending on the cell types/tumor cell content having been sampled. Am I correct to assume that about half of the cancer-associated proteins the authors had previously assembled ($n=349$) were not quantified in the current study ($n=154$). Are the missing proteins not expressed in the tissue/cell types analyzed or are they not detected? Datasets, like CPTAC and their own RNA-seq data might provide clues.

We thank the reviewer for this excellent suggestion. We have now added additional analyses based on our RNA-seq data (new Fig. EV1A,C,D, Appendix Fig. S2C-F). As expected, about half of the genome was quantified by RNAseq and considered expressed. We quantify proteins for about half of the expressed genes. The same applies for the cancer-associated proteins where about half are not detected at protein level. While it is possible to increase the fraction of quantified proteins to the order of 10^5 - 10^6 this comes at a high cost. It requires either extensive time-consuming involving fractionation or high capital investment for top-of-the-line instrumentation. Both dramatically increase the cost of the analysis and are a barrier to wide spread implementation. Our study has shown that proteome guided precision oncology is feasible on older generation or lower performance instrumentation with reasonable instrument time provides sufficient coverage of the proteome to obtain robust actionable insights.

The authors determined IC50 values for SETTLE cells and found these to be between 5.5 and 10.1 μM (Fig. 4C). How do these IC50 values compare to drug effects in normal (lung) tissue/mice? What are potential side effects of the drug at this concentration? 'Sertraline HCl (32 $\mu\text{mol/kg}$ i.p.) inhibits serotonin uptake into striatal synaptosomes from rats by more than 50%' Selleckchem (<https://www.selleckchem.com/datasheet/sertraline-hcl-S405302-DataSheet.html>). The authors should compare their experimentally identified IC50 values with some reference.

The sertraline IC50 value was determined using SETTLE tumour cells (the source was dissociated and mouse cell-depleted NSG PDX tumor of R2)(Fig. EV3C). In a recent review of sertraline effects as an anti-tumor agent (PMID:36291722), IC50 values of between ~ 1 - $15 \mu\text{M}$ was reported for cancers of the lung, colorectal, breast, liver and blood, which is comparable to what we have observed for SETTLE. For the *in vivo* CAM and zebrafish experiments (Fig. 3B; Fig. EV3D,E), the quantity of sertraline administered was always within the pre-established maximum tolerated dose for each model system, and referencing the respective experimental norms for murine PDXs (PMID:33203732; 1-2 mg/kg for CAM), and 2-4 μM for zebrafish.

The authors use the heading: "Sertraline and trimethoprim combination as a potential therapeutic approach for SETTLE tumors". The authors analyzed one patient. Can they indeed generalize for SETTLE as an entity? Their findings and rationale may be very relevant for other patients and entities that rely on 1C carbon metabolism - the authors cite (ref. 37) a study having tested serine-dependent cell models of breast cancer, and present drug-synergy data having been obtained using two TNBC cell lines (this could become kind of a third story).

We fully agree with the reviewer that this statement made a generalization that was not backed by data. While we now have confirmed elevation of SHMT2 in 5 additional settle cases (see discussed in response to reviewer 1) we have no viable material from these patients and cannot assess if our statement holds. We therefore rewrote this section to reflect the n of 1 nature of the study more clearly.

In Fig. 6D the decline of monthly growth rates is shown. Fig. 6D gives percentages/month with mostly steep declines upon setraline therapy (pink) compare to pre setraline (black), and further decreases post setraline (yellow). How were monthly tumor growth rates assessed prior to setraline treatment?

We appreciate pointing out that this benefits from additional clarity. We have two CT scans that were taken before sertraline therapy. The tumor growth prior to sertraline treatment is calculated as the change in tumor size between these two timepoints. To communicate this more clearly, we have now separated the CT scans and growth curves in two tiles and in both labeled the three timespans as A: before sertraline treatment; B: on sertraline treatment; C after sertraline treatment (Fig. 3F,G).

Minor:

In the introduction, the authors might cite some study that has described the outcome specifically for pediatric patients, e.g., PMID:34492587. It should be noted that those improvements have mostly not been achieved based on genomic profiling though. The current challenge in pediatric oncology is relapsed and metastatic disease as those tumors are mostly refractory to therapy.

We thank the reviewer for this suggestion and have updated and expanded the introduction to incorporate this.

The authors state that INFORM has not reported "on the response or survival status [of patients] (13)." The authors might consider PMID:37364231

We thank the reviewer for pointing this out and have corrected the introduction accordingly.

Still in the introduction, the authors write about "challenges arising from the limited quantity and of available viably cryopreserved tissues ... (15)." The authors should be aware that their reference 15 only lists studies that used fresh tissues for generation of PDX models. Cryopreserved tissue has rarely been reported for generation of PDX models (e.g., PMID: 33910584). Utilization of fresh tissues might speed up engraftment of PDXs.

We appreciate the reviewer pointing out that the reference was not placed accurately and that the statement lacked clarity. We have now updated the respective passage and cited PMID:33910584, a study that compared the efficacy of PDX generation from cryopreserved or fresh tumor tissues.

Figure 1D lists 29 proteins, not 14. It should be explained that the remaining proteins did not map to respective interventional baskets.

Thank you for pointing out this inconsistency. We have now changed the text and figure legend to clarify that only associations for quantified proteins are shown.

The authors found significant increases in PTEN levels. Increased abundance of PTEN might be associated with enhanced growth factor PI3K-mTor signaling activity in the tumor compared to adjacent normal lung tissue. An assessment of pathway activities (e.g., using RNA-seq data) might be warranted.

This is an excellent suggestion. We have now performed this analysis based on our RNAseq data and expanded the result section as follows:

"The only significant change was increased PTEN protein levels in tumor nodules linked to a tumor suppressive effect (26). Alongside PTEN, the PI3K-AKT-mTOR pathway including PIK3CA, AKT2, AKT3, and MTOR showed high RNA expression (Appendix Table S1) but with no significant protein level changes and thus, was considered not clinically actionable by the MTB."

The authors write that they were able to rapidly generate CAM-PDXs [which] afforded an opportunity to rapidly evaluate targeting the 1C metabolic pathway. How rapid was generation and testing of this CAM-PDX model?

Thank you for the opportunity to clarify. As detailed above the CAM-PDXs can be generated from viably cryopreserved SETTLE tumors in only one cycle of xenografting and assayed 5-7 days post implantation. The drug assays were completed within 4 cycles of xenografting. Please see the earlier response to the question for details

regarding the establishment of PDXs.

Referee #3 (Comments on Novelty/Model System for Author):

The results with the CAM xenograft model were difficult to interpret from the figures shown and the response to the proposed targeted therapy was modest. Since this was an 'n of one' study, it is impossible to tell if the model was insufficiently sensitive, or if the tumor was heterogeneous and difficult to model. These points are in my Comments to Authors

We appreciate the reviewer's careful evaluation of our xenograft data and have carefully revised the text and figures to address the critical aspects of model sensitivity and tumor heterogeneity. We provide a detailed account of this in our response to the specific comments below.

Referee #3 (Remarks for Author):

The manuscript 'Proteomics and personalized patient-derived xenograft models identify treatment opportunities for a progressive malignancy within a clinically actionable timeframe and change care' by Barnabas et al is a very interesting and potentially consequential demonstration of two important points: 1) that adding measurements of proteins (proteomics) to traditional 'precision oncology' measurements of DNA and RNA can contribute significant new insights to individualized therapeutic interventions, and 2) that adding proteomic measurements is clinically feasible within a two week time frame. Additionally, they propose a workflow for establishing a patient-derived xenograft in a chick allantoic membrane (CAM) model system that can provide ex vivo results to a Molecular Tumor Board within a two month time period.

While the potential power of adding proteomics and ex vivo testing on patient derived xenografts in the context of precision oncology is truly exciting and highly transformative, the current work has several significant limitations. This is an 'n of one' study, with a single patient, so no comparison can be drawn between this approach and the more traditional nucleic acids only approaches.

Because it is an n of one study and patient-derived materials are seriously limited, insufficient evidence is presented to truly validate the CAM xenograft system as an adequate model system for identifying individualized therapeutic strategies. The experimental evidence provided to support ex vivo testing is weak, and it is impossible to tell whether that is because the model system has limited efficacy or because the n of one patient simply had a highly heterogeneous and treatment resistant tumor by the time that proteomics was introduced, at the third recurrence as lung metastases.

The authors are well aware of the limitations imposed by the 'n of one' design, but they justify submitting this preliminary result in order to stimulate interest in the approach and encourage other investigators with access to appropriate patients and clinical trials to attempt the same approach. This is a highly laudable motive, particularly as that type of multi-center, multi-investigator, multi-disease data is precisely what will be needed to either confirm or reject the clinical utility of adding proteomics to target identification and CAM xenografts to target validation in a practical clinical context. That said, there are some changes that could be made to the current manuscript to improve its usefulness to other researchers in the field and help avoid enthusiasm beyond what has actually been demonstrated.

We are pleased to hear that the reviewer considers the potential of adding proteomics and ex vivo testing to precision oncology, as demonstrated in our study, as truly exciting and informative and agrees with us that its publication, even at the early stage of an N-of-one study, is important.

As the reviewer points out, we fully acknowledge the limitations of our N-of-one study, we would however like to point out that the study provides an internal comparison to traditional nucleic acid only approaches as these were performed in parallel and did not yield any further actionable targets for this patient at this disease stage. As requested by the editor and reviewers we have now rewritten the manuscript as a case report to further acknowledge the N-of-one nature of the study.

Suggested Improvements:

1. In reading the text linearly, there were several questions critical to evaluation of the results that were not addressed in the text, but were handled nicely in the Figures. The text should either address these questions directly or clearly direct the reader to the appropriate figure for the following:

a. The precise timeline of initial nucleic acid testing, administration of nucleic acid-based targeted therapies, the various relapses, and the addition of proteomic measurements over the course of the patient's disease is unclear in the text. All of that is handled very nicely in Figure 1B. However, the figure legend for 1B should be explicit that the proteomic analysis of PR and R1 was retrospective.

Similarly, Figure 1A does a great job of delineating the time required for each step compared to the standard clinical evaluation process; this could have been clarified in the text.

We have added a more detailed and accessible account of the timeline to the text and made the suggested changes to the legend. We thank the reviewer for their suggestion and find the new version much improved. We also flipped the order of Fig 1A and B, which aligns better with the shortened version of this resubmission. The respective section in the results and figure legend now reads:

“At initial diagnosis the child had presented with asymptomatic right neck swelling, which had been incidentally noted. Ultrasound showed a 3.8 x 2.3 x 4.5cm heterogeneous soft tissue mass replacing the right thyroid lobe, with no associated adenopathy. The patient underwent right hemithyroidectomy confirming a SETTLE tumor (Matheson et al, 2019) (Appendix Text – Patient History). The patient was enrolled in the PROFYLE study when they experienced a first recurrence or relapse (R1) with metastatic progressive disease (Fig. 1A).”

At R1, we conducted whole genome and transcriptome profiling and the patient received Sorafenib (Appendix Text – R1 Genome Profiling). At R3, we conducted prospective real-time LC-MS/MS-based quantitative whole proteome profiling, candidate target identification, and IHC target validation within 2 weeks of biopsy. To expand the molecular etiology, we also conducted retrospective proteome profiling of biopsies from the primary tumor and recurrences R1 and R2. We used viably cryopreserved R2 tissue to establish and evaluate drug response as CAM PDXs. In parallel, R2 tumor was also established as mouse PDXs, affording expansion and archiving of SETTLE cells for additional CAM and zebrafish ex vivo drug studies, and to confirm on-target activity by isotope tracing in vitro. Whole exome/transcriptome sequencing was performed at R3 to identify actionable alterations not present at R1. Two months after biopsy the combined pre-clinical insights were evaluated by the PROFYLE Molecular Tumor Board (MTB) followed by initiation of innovative therapy (Fig. 1A,B).”

“Figure 1: Multi-omics molecular profiling of a progressive SETTLE case

A. Patient journey from the resection of the tumor (PR) through three distant recurrences (R1, R2, R3). The molecular analyses, radiographs (Months at time X, MX to M+7), PDX models, and drug sensitivity assays conducted are linked to the corresponding time points. Proteome analysis of PR, R1 and R2 was performed retrospectively at time of R3.

B. Timeline of precision diagnostics for the SETTLE disease course, including multi-omics molecular tumor profiling, real-time target identification, and validation using personalized xenograft models in providing timely pre-clinical support for medical decision-making.”

2. When describing the precision oncology analysis of the R1 tumor, please specify that the 'increased expression' observed for specific genes was at the RNA level. Proteins are gene products, too, and not all readers will assume 'gene expression' is automatically mRNA.

The text have been revised as suggested.

3. Since the investigators did a retrospective proteomic analysis of tumor from R1, which also has genomics and transcriptomics, a table comparing the changes observed in the R1 tumor at the DNA, RNA, and protein levels would have been informative in evaluating what is added by proteomics. This could be done by adding proteins to Supplemental Table 1 or by constructing a new table that highlighted only the changes.

We thank the reviewer for this excellent suggestion and have incorporated it in the revised Supplementary Table 1.

4. The quality of the proteomic data shown is good, but given the only 44% overlap between the DIA results and known 'pediatric cancer proteins' in Fig 1F, information on the subcellular distribution of the identified proteins would have been helpful, specifically, whether there was enrichment of cytoplasmic versus membrane proteins. Suppl Figure 1C shows biological processes, but not subcellular localization.

The reviewer raises an excellent point and we have conducted several additional analyses based on this suggestion as well as the suggestion of reviewer 1. The subcellular distribution of quantified proteins shows balanced coverage across all compartments including membrane proteins. In line with the distributions seen in a normal shotgun proteomic experiment cytoplasmic proteins show some enrichment and membrane proteins a somewhat underrepresented.

We have added this data as Appendix Fig. S2A and added the following details to the Appendix Text:

“Evaluation of subcellular location annotations for quantified proteins showed the typical moderate derichment of nucleoplasm, vesicular, and plasma membrane locations and enrichment of proteins annotated to be localized to the cytoplasm, mitochondria, and endoplasmic reticulum with overall balanced coverage across all subcellular locations (Fig. S2A).”

As detailed above we have also added additional analysis to determine what fraction of the genome, and in particular the 'pediatric cancer proteins' is expressed at the RNA level using our RNAseq data. This shows that we capture 52% of pediatric cancer proteins at expressed at R1 and 47% of those expressed at R3.

5. Were any PTMs identified in the proteomic data, particularly phosphopeptides? Given the somewhat lower than expected representation of kinases in the observed known proteins, this could be relevant to the ability to identify potential targets.

We agree with the point raised by the reviewer that our analysis may have missed relevant targetable kinases. We did perform additional analysis focussing on the abundance receptor tyrosine kinases to investigate the emerging resistance to Sorafenib showing that we quantify many key members of this group. We did not have sufficient biopsy material to enrich for phosphopeptides and faithfully assess phosphorylation and the activation status of kinases. We envision inclusion of this in future studies as this becomes more feasible on newer generation mass spectrometers with increased sensitivity. To communicate this more clearly, we have added new Appendix Text and Fig. S1A and S1B:

“Several receptor tyrosine kinases showed reduced RNA expression from R1 to R3 (Fig S1A), and RNA and protein levels of FLT3 were low or below the detection limit at R1 and R3 (Fig S1B, Table S1), potentially explaining the acquired resistance to Sorafenib. As the biopsy material was too limited, we could not perform phosphopeptide enrichment to assess kinase and pathway activation status. We envision including this in future studies to expand the ability to characterize this important target group.”

6. It is really difficult for the reader to interpret the micrographs in Figure 4A. The reflections off the xenograft and the differences in lighting are truly distracting, and it is not even clear what is the tumor in the D1 sertraline image. In comparison, the micrographs in Suppl Figure 5A are much easier to interpret. Arrows should be used to define the margins of the xenograft. Better images would help. I had to look up a review article on CAM xenografts to interpret what I was seeing.

Our apologies for the choice of representative micrographs that weren't always consistent in terms of the lighting effects, the placement of the tumor xenografts relative to the yolk and embryo, and, capturing a 'still-frame' in between the cardiac oscillations (rhythmic heartbeat) and other movements of the live chick embryo. We have updated the micrographs shown (now Fig. 3A) and included markings/tracings to better highlight the tumoroid boundaries and measurements for assessing drug response in the CAM PDX model. We have replaced the Veh Ctrl set of images with ones that have less reflections or glare. The longitudinal image set as shown is for the same xenografted microtumor over the 5 days of assay. In consolidating the manuscript to better align the presentation of this Case Report, the CAM-PDX drug data is now shown for sertraline and SHIN1 treatment as Fig. 3B and Fig. 3C, respectively. We will include the full resolution images for each panel presented as part of the submission of source data as recommended by the journal/publisher.

7. One is left with the overall impression that the utility of the CAM xenograft model is somewhat overstated in the manuscript. The CAM xenograft histology in Figure 3C (now Fig 2F) is very different from both the patient's R2 (Figure 3A; now Fig 2C but not the HnE) and NSG-PDX (Figure 3B; now Fig 2E); interestingly, the image in Suppl Fig 3A (now Fig S5) is much more convincing.

The morphology of the individual spindle cells seen in the CAM tumours show features that are identical to isolated tumor cells in discohesive areas of the primary and recurrent tumours, and they were validated with a human marker (LAMP1). Solid fascicular growth pattern was not observed in the CAM-PDX samples, for which we are unsure of the etiology but may relate to either not enough time to grow into this pattern (5-7 days for CAM-PDX), compared to the mouse (6-8 months for the initial NGS-PDX), or may be an idiosyncratic reason (SETTLE being a slow growing indolent tumor). The morphology seen in the supplement may look more like the primary and relapse samples due to the cellularity; however, the cells still show individual growth with no well formed fascicular pattern.

7. (ctd...) A more honest assessment of the preliminary nature of the ex vivo drug testing in the CAM-PDX model with some suggestions for improvement would have bolstered confidence in the potential clinical utility of this workflow. Given the modest effects of sertraline in the CAM-PDX data shown, it is not at all surprising that the patient only showed a reduction in tumor growth rate in response to sertraline treatment. Had there been an example of a more impressive ex vivo response correlated with a more significant clinical outcome, it actually would have enhanced confidence in the ability of the CAM-PDX system to recapitulate the patient's tumor biology.

We appreciate the reviewer pointing out that the reader might misinterpret our statements on combination treatments. We very much agree that the sertraline/trimethoprim combination treatment (Fig. EV3D,E,F) showed no synergy, and have revised our statement to better reflect this. Our intent was to show that trimethoprim or sertraline alone showed inhibitory effects when assessed as CAM and zebrafish PDXs (Fig. EV3D,E), even if trimethoprim alone exhibited no measurable response in *in vitro* assays (Fig. EV3F) of the SETTLE cells.

It is meaningful to point out that the combination studies (Fig. EV3D,E,F) were conducted with SETTLE tumor cells that were expanded as NSG PDXs (which took between 6-8 months). This was post the MTB discussion, which had only considered the genomics, proteomics and CAM PDX testing data with sertraline alone (Fig. 3A,B). We also agree that the effects of sertraline evaluated at Day 3 is not significant when compared to the Vehicle Control. However, inclusion of Day 3 data provides context to the treatments administered, which in this case was an initial dose at Day 1 and a second dose at Day 3.

We agree that the *ex vivo* drug testing using the CAM or zebrafish models on the SETTLE tumor is preliminary in nature and lacks the target pathway specificity (eg targeting SHMT2 with sertraline or SHIN1). The limited available patient derived SETTLE tissue allowed for only the pre-clinical assays we have shown here, hence few other agents could be evaluated on this SETTLE tumor.

To address the concern of demonstrating a positive or negative response using the CAM-PDX model, we have now included sertraline drug testing of CAM xenografts of MDA-MB-468 and MDA-MB-231, respectively, breast tumor cell lines previously demonstrated to be serine-addicted and non-addicted (PMID:33203732). The newly added data (Fig. EV3B) provides supporting evidence that the serine-addicted MDA-MB-468 tumors exhibited enhanced sensitivity (or positive response) to sertraline. To confirm the specificity of the link between the elevated 1C metabolism and sertraline treatment, we had conducted stable isotope tracing experiments that clearly showed the on-target effect of the treatment (Fig.3 D,E) on SETTLE cells. We have now revised the text to communicate this more clearly.

We thank all three reviewers for their thoughtful comments and critical review that helped us to re-submit a much improved and streamlined manuscript that includes new data and addresses all points raised.

17th Feb 2025

Dear Dr. Lange,

Thank you for the submission of your revised manuscript to EMBO Molecular Medicine. I am pleased to inform you that we will be able to accept your manuscript pending the following final amendments:

- 1) Please address referee #3 minor points.
- 2) Authors: We note name discrepancy in the manuscript and in our submission system: Andy J. Mungall vs. Andrew J Mungall. Please correct. Also, please provide corr. author information on the title page of the manuscript.
- 3) In the main manuscript file, please do the following:
 - Please address all comments suggested by our data editors listed below:
 - o Figure legends:
 1. Please note that the exact p values are not provided in the legends of figures 2B, 3B, C, G; EV1 A, C, D; EV3 B, D, E; supplementary figures 2D, G.
 2. Please indicate the statistical test used for data analysis in the legends of figures 2B, 3B, C, G; EV2 A, EV3 B, D, E; supplementary figures 2A, D, G.
 3. Please indicate what */ **/ ***/ **** represents; if this represents p value(s), please indicate the statistical test used and where appropriate, specify the exact p value in the legend(s) of figure(s) EV2 A, B."
 4. Please note that the box plots need to be defined in terms of minima, maxima, centre, bounds of box and whiskers, and percentile in the legends of supplementary figures 1C.
 5. Please note that information related to n is missing in the legends of figures 1D, 2B, 3B, C, D, E; EV2 A, B; EV3 B, C, D, E; supplementary figures 1C, 2A, D-G.
 6. Please note that the error bars are not defined in the legends of figures 3D, E; EV2 A, B; EV3 C, supplementary figures 2D-G.
 - Limit keywords to max. 5.
 - Rename Extended Figure 1 etc. to Figure EV1 etc.
 - Rename "Conflict of interests" to "Disclosure Statement & Competing Interests". We updated our journal's competing interests policy in January 2022 and request authors to consider both actual and perceived competing interests. Please review the policy <https://www.embopress.org/competing-interests> and update your competing interests if necessary.
 - Indicate in legends exact n and exact p values, not a range, along with the statistical test used. To keep the figures "clear" some authors found providing an Appendix table Sx with all exact p-values preferable. You are welcome to do this if you want to.
 - Please remove Reagents and Tools Table and uploaded it as a separate file. Structured Methods section includes Reagents and Tools Table followed by a Methods and Protocols section. More information on how to adhere to this format as well as downloadable templates (.docx) for the Reagents and Tools Table can be found in our author guidelines: <https://www.embopress.org/page/journal/17574684/authorguide#structuredmethods>
An example of a paper with Structured Methods can be found here: <https://www.embopress.org/doi/full/10.1038/s44320-024-00037-6#sec-4>
 - 4) Appendix: Please remove appendix Patient history and fuse it with the one in the results as they are largely redundant. Please group the rest of appendix text in the section named "Appendix Supplementary Information" and update the callouts in the main manuscript text. Please rename appendix tables and figures to Appendix Table S1 and Appendix Figure S1 etc. and update their callouts in the main manuscript file.
 - 5) Funding: Please make sure that information about all sources of funding are complete in both our submission system and in the manuscript. All sources of funding with the grant number listed in the Acknowledgement should be entered separately in our submission system.
 - 6) The Paper Explained: Please provide "The Paper Explained" and add it to the main manuscript text. Please check "Author Guidelines" for more information. <https://www.embopress.org/page/journal/17574684/authorguide#researcharticleguide>
 - 7) Synopsis: Every published paper now includes a 'Synopsis' to further enhance discoverability. Synopses are displayed on the journal webpage and are freely accessible to all readers. They include separate synopsis image and synopsis text.
 - Synopsis image: Please provide a striking image or visual abstract as a high-resolution jpeg file 550 px-wide x (300-600)-px high to illustrate your article.
 - Synopsis text: Please provide a short standfirst (maximum of 300 characters, including space) as well as 2-5 one sentence bullet points that summarise the paper as a .doc file. Please write the bullet points to summarise the key NEW findings. They should be designed to be complementary to the abstract - i.e. not repeat the same text. We encourage inclusion of key acronyms and quantitative information (maximum of 30 words / bullet point). Please use the passive voice.
 - Please check your synopsis text and image before submission with your revised manuscript. Please be aware that in the proof stage minor corrections only are allowed (e.g., typos).
 - 8) Source data: Please upload EV figure source data a single zip folder.
 - 9) As part of the EMBO Publications transparent editorial process initiative (see our Editorial at <http://embomolmed.embopress.org/content/2/9/329>), EMBO Molecular Medicine will publish online a Review Process File (RPF) to accompany accepted manuscripts. This file will be published in conjunction with your paper and will include the anonymous referee reports, your point-by-point response and all pertinent correspondence relating to the manuscript. Let us know whether you agree with the publication of the RPF and as here, if you want to remove or not any figures from it prior to publication.

10) Please provide a point-by-point letter INCLUDING my comments as well as the reviewer's reports and your detailed responses (as Word file).

I look forward to reading a new revised version of your manuscript as soon as possible.

Yours sincerely,

Zeljko Durdevic

*** Instructions to submit your revised manuscript ***

- 1) a .docx formatted version of the manuscript text (including Figure legends and tables)
- 2) Separate figure files*
- 3) supplemental information as Expanded View and/or Appendix. Please carefully check the authors guidelines for formatting Expanded view and Appendix figures and tables at <https://www.embopress.org/page/journal/17574684/authorguide#expandedview>
- 4) a letter INCLUDING the reviewer's reports and your detailed responses to their comments (as Word file).
- 5) The paper explained: EMBO Molecular Medicine articles are accompanied by a summary of the articles to emphasize the major findings in the paper and their medical implications for the non-specialist reader. Please provide a draft summary of your article highlighting
 - the medical issue you are addressing,
 - the results obtained and
 - their clinical impact.This may be edited to ensure that readers understand the significance and context of the research. Please refer to any of our published articles for an example.
- 6) Author contributions: the contribution of every author must be detailed in a separate section.
- 7) EMBO Molecular Medicine now requires a complete author checklist (<https://www.embopress.org/page/journal/17574684/authorguide>) to be submitted with all revised manuscripts. Please use the checklist as guideline for the sort of information we need WITHIN the manuscript. The checklist should only be filled with page numbers where the information can be found. This is particularly important for animal reporting, antibody dilutions (missing) and exact values and n that should be indicated instead of a range.

8) Every published paper now includes a 'Synopsis' to further enhance discoverability. Synopses are displayed on the journal webpage and are freely accessible to all readers. They include a short stand first (maximum of 300 characters, including space) as well as 2-5 one sentence bullet points that summarise the paper. Please write the bullet points to summarise the key NEW findings. They should be designed to be complementary to the abstract - i.e. not repeat the same text. We encourage inclusion of key acronyms and quantitative information (maximum of 30 words / bullet point). Please use the passive voice. Please attach these in a separate file or send them by email, we will incorporate them accordingly.

You are also welcome to suggest a striking image or visual abstract to illustrate your article. If you do please provide a jpeg file 550 px-wide x 300-600px high.

9) A Conflict of Interest statement should be provided in the main text

10) Please note that we now mandate that all corresponding authors list an ORCID digital identifier. This takes <90 seconds to complete. We encourage all authors to supply an ORCID identifier, which will be linked to their name for unambiguous name identification.

Currently, our records indicate that the ORCID for your account is 0000-0003-1171-5864.

Link Not Available

11) Include a Reagents and Tools Table as part of the Methods section, which can be downloaded from our author guidelines (<https://www.embopress.org/page/journal/17574684/authorguide#structuredmethods>)

Photos 400-800 DPI

*Additional important information regarding figures and illustrations can be found at

<https://bit.ly/EMBOPressFigurePreparationGuideline>. See also figure legend preparation guidelines:

<https://www.embopress.org/page/journal/17574684/authorguide#figureformat>

***** Reviewer's comments *****

Referee #1 (Comments on Novelty/Model System for Author):

As was indicated in my original review, the authors have gone to quite extraordinary lengths to establish their approach in this case report. The technical quality of the experiments was high, and has been enhanced with some of the additional controls included in the rebuttal. The medium medical impact reflects the fact that this work has been revised to more closely resemble a case report in which the multi-omics methodologies were applied. This is now, in my view, a more appropriate way to present this data

Referee #1 (Remarks for Author):

The authors have addressed the bulk of the comments and suggestions raised in my original review. I believe the re-framing of this work as a case study is appropriate.

Referee #2 (Comments on Novelty/Model System for Author):

The authors now focus on the very thorough description of a single case. The depth of their investigation makes this a strong study. The medical impact is not fully clear based on this study alone, however, it can be assumed that proteomics analysis

combined with experimental (drug) testing will be the next step in precision oncology. There, the authors make some strong points about particular limitations and benefits that will likely impact also other studies.

Referee #2 (Remarks for Author):

The authors have carefully responded to the reviewers' points. This is a highly relevant study and I look forward to reading about the first results from expanding this to children and adolescents with hard-to-treat cancers across Canada.

Referee #3 (Comments on Novelty/Model System for Author):

Ranking the medical impact is difficult for this case study. In terms of impact on the case study patient, the response was only moderate (a slowing of tumor growth rate compared to extrapolation of pre-treatment growth). However, the POTENTIAL medical impact would be very high, if this manuscript, and subsequent efforts to refine and expand the real time addition of proteomics data to the precision oncology workflow, leads to the routine use of protein data in addition to DNA and RNA data, the impact will be huge. Viewed as an imperfect but necessary first step, this manuscript deserves to be published.

Referee #3 (Remarks for Author):

The resubmission of the manuscript 'Proteomics and personalized PDX models identify treatment for a progressive malignancy within an actionable timeframe' is very significantly improved in comparison to the prior version. By re-casting the manuscript as a 'case study', the authors emphasize the 'n of 1' nature of the study and provide a strong acknowledgement of the weaknesses inherent in a case study of one patient. Still, they manage to retain the most significant strength of the manuscript, which is the simple demonstration that proteomic analysis can be added to traditional genomic and transcriptomic analyses on a time frame compatible with the decision process of Molecular Tumor Boards. While the limited clinical response to sertraline therapy, the novel therapeutic strategy derived from the proteomic analysis, is disappointing, this is in the setting of a tumor that had recurred three times despite targeted anti-FLT3 therapy guided by traditional precision oncology approaches based on genomics. Moreover, there were no changes evident in the transcriptomic data to indicate another line of treatment, other than to explain the resistance to RTK inhibitors. Clearly this was a difficult to treat tumor, and one would hope that expansion of this general approach would lead to more significant clinical responses - a hypothesis that can only be tested if additional research centers are persuaded to attempt this workflow, based on the feasibility demonstration in this report.

The most significant improvement to the manuscript in its current form is the clarification of the actual workflow used, in particular the timeline between tumor excision, proteomic analysis, transcriptome analysis, CAM model establishment, and drug testing. The new text is perfectly clear and aided by the improvements to Figure 1. All my concerns were addressed in those changes. Since the other two reviewers had difficulty following the timeline as well, leading to some confusion regarding the intent and utility of the study, the timeline clarification should help other readers better understand the intent of the exercise. The additional cell line experiments added in response to Reviewer #1 are also a significant improvement, as is the expansion of the tumor-normal comparisons to include public data from other tumors and tumor types, provided in response to Reviewer #2.

This reviewer raised several concerns about the CAM model, and particularly its representation in original Figure 3; the new version of Figure 3 is much improved and the technical questions have been adequately addressed. The subsequent validation of the CAM results in murine PDX models and zebrafish is encouraging, and the justification of the mouse PDX as a necessary intermediary for expanding the amount of human tumor tissue available is reasonable. Other technical issues raised by reviewer #2 regarding the use of patient-derived organoid models (limited by the lack of protocols for SETTLE tumors), the use of FFPE for proteomics and viable cryopreserved specimens for in vivo models, and the use of IHC to validate SHMT2 overexpression in this and other patients appear to be adequately addressed in this revision. There is one very minor point: the correlation values between RNA and protein expression levels of 0.42-0.43 discussed on page 5 and shown in Extended Figure 1 (panels C and D) are not really 'low correlations', but actually completely in line with multiple analyses of RNASeq and MS data from the same tumor, as demonstrated in numerous CPTAC publications. A simple reference would do.

23rd Feb 2025

Dear Dr. Zeljko Durdevic,

We thank you and the three reviewers for having considered our revised manuscript EMM-2024-20488-V2 entitled "Proteomics and personalized PDX models identify treatment for a progressive malignancy within an actionable timeframe". We appreciate your decision to accept the manuscript with further amendments, as detailed below in the Point by Point Response.

Point by Point Response to *Reviewers' and Editor's Comments*

Referee #1 (Comments on Novelty/Model System for Author):

As was indicated in my original review, the authors have gone to quite extraordinary lengths to establish their approach in this case report. The technical quality of the experiments was high, and has been enhanced with some of the additional controls included in the rebuttal. The medium medical impact reflects the fact that this work has been revised to more closely resemble a case report in which the multi-omics methodologies were applied. This is now, in my view, a more appropriate way to present this data

Referee #1 (Remarks for Author):

The authors have addressed the bulk of the comments and suggestions raised in my original review. I believe the re-framing of this work as a case study is appropriate.

Response: We thank the reviewer for their thoughtful and thorough consideration of our original and revised manuscript.

Referee #2 (Comments on Novelty/Model System for Author):

The authors now focus on the very thorough description of a single case. The depth of their investigation makes this a strong study. The medical impact is not fully clear based on this study alone, however, it can be assumed that proteomics analysis combined with experimental (drug) testing will be the next step in precision oncology. There, the authors make some strong points about particular limitations and benefits that will likely impact also other studies.

Referee #2 (Remarks for Author):

The authors have carefully responded to the reviewers' points. This is a highly relevant study and I look forward to reading about the first results from expanding this to children and adolescents with hard-to-treat cancers across Canada.

Response: We thank the reviewer for their thoughtful and thorough consideration of our original and revised manuscript.

Referee #3 (Comments on Novelty/Model System for Author):

Ranking the medical impact is difficult for this case study. In terms of impact on the case study patient, the response was only moderate (a slowing of tumor growth rate compared to extrapolation of pre-treatment growth. However, the POTENTIAL medical impact would be very high, if this manuscript, and subsequent efforts to refine and expand the real time addition of proteomics data to the precision oncology workflow, leads to the routine use of protein data in addition to DAN and RNA data, the impact will be huge. Viewed as an imperfect but necessary first step, this manuscript deserves to be published

Referee #3 (Remarks for Author):

The resubmission of the manuscript 'Proteomics and personalized PDX models identify treatment for a progressive malignancy within an actionable timeframe' is very significantly improved in comparison to the prior version. By re-casting the manuscript as a 'case study', the authors emphasize the 'n of 1' nature of the study and provide a strong acknowledgement of the weaknesses inherent in a case study of one patient. Still, they manage to retain the most significant strength of the manuscript, which is the simple demonstration that proteomic analysis can be added to traditional genomic and transcriptomic analyses on a time frame compatible with the decision process of Molecular Tumor Boards. While the limited clinical response to sertraline therapy, the novel therapeutic strategy derived from the proteomic analysis, is disappointing, this is in the setting of a tumor that had recurred three times despite targeted anti-FLT3 therapy guided by traditional precision oncology approaches based on genomics. Moreover, there were no changes evident in the transcriptomic data to indicate another line of treatment, other than to explain the resistance to RTK inhibitors. Clearly this was a difficult to treat tumor, and one would hope that expansion of this general approach would lead to more significant clinical responses - a hypothesis that can only be tested if additional research centers are persuaded to attempt this workflow, based on the feasibility demonstration in this report.

The most significant improvement to the manuscript in its current form is the clarification of the actual workflow used, in particular the timeline between tumor excision, proteomic analysis, transcriptome analysis, CAM model establishment, and drug testing. The new text is perfectly clear and aided by the improvements to Figure 1. All my concerns were addressed in those changes. Since the other two reviewers had difficulty following the timeline as well, leading to some confusion regarding the intent and utility of the study, the timeline clarification should help other readers better understand the intent of the exercise. The additional cell line experiments added in response to Reviewer #1 are also a significant improvement, as is the expansion of the tumor-normal comparisons to include public data from other tumors and tumor types, provided in response to Reviewer #2.

This reviewer raised several concerns about the CAM model, and particularly its representation in original Figure 3; the new version of Figure 3 is much improved and the technical questions have been adequately addressed. The subsequent validation of the CAM results in murine PDX models and zebrafish is encouraging, and the justification of the mouse PDX as a necessary intermediary for expanding the amount of human tumor tissue available is reasonable. Other

technical issues raised by reviewer #2 regarding the use of patient-derived organoid models (limited by the lack of protocols for SETTLE tumors), the use of FFPE for proteomics and viable cryopreserved specimens for in vivo models, and the use of IHC to validate SHMT2 overexpression in this and other patients appear to be adequately addressed in this revision. There is one very minor point: the correlation values between RNA and protein expression levels of 0.42-0.43 discussed on page 5 and shown in Extended Figure 1 (panels C and D) are not really 'low correlations', but actually completely in line with multiple analyses of RNASeq and MS data from the same tumor, as demonstrated in numerous CPTAC publications. A simple reference would do.

Response: We thank the reviewer for their thoughtful and thorough consideration of our original and revised manuscript. We appreciate the suggestion regarding the RNASeq and MS data correlation, and have rephrased accordingly and with citation of the CPTAC studies. As follows: “...and correlation of 0.43 (R1) and 0.42 (R3) with transcript levels that are consistent with RNASeq and MS data analyses from the same tumor (**Fig. EV1C,D**), as shown in CPTAC studies (Savage *et al*, 2024; Zhang *et al*, 2022).”

Editor’s Decision Letter

Thank you for the submission of your revised manuscript to EMBO Molecular Medicine. I am pleased to inform you that we will be able to accept your manuscript pending the following final amendments:

1) Please address referee #3 minor points.

Addressed accordingly with revision of the sentence and inclusion of additional citations.

2) Authors: We note name discrepancy in the manuscript and in our submission system: Andy J. Mungall vs. Andrew J Mungall. Please correct. Also, please provide corr. author information on the title page of the manuscript.

Now revised to the Andrew J Mungall. Corresponding authors’ email contact information has been provided.

3) In the main manuscript file, please do the following:

- Please address all comments suggested by our data editors listed below:

o Figure legends:

1. Please note that the exact p values are not provided in the legends of figures 2B, 3B, C, G; EV1 A, C, D; EV3 B, D, E; supplementary figures 2D, G.

Figure legend revised to include P values for figures 2B; 3B,C,G; EV1 A,C,D; EV3 B,D,E; and S2D,G.

2. Please indicate the statistical test used for data analysis in the legends of figures 2B, 3B, C, G; EV2 A, EV3 B, D, E; supplementary figures 2A, D, G.

Figure legend revised to include statistical tests for figures 2B; 3B,C,G; EV2 A; EV3 B,D,E; and S2A,D,G.

*3. Please indicate what */ **/ ***/ **** represents; if this represents p value(s), please indicate the statistical test used and where appropriate, specify the exact p value in the legend(s) of figure(s) EV2 A, B."*

Figure legend revised with p values for each */**/***/****

4. Please note that the box plots need to be defined in terms of minima, maxima, centre, bounds of box and whiskers, and percentile in the legends of supplementary figures 1C.

Box plot information now appended to figure legend of Figure S1C

5. Please note that information related to n is missing in the legends of figures 1D, 2B, 3B, C, D, E; EV2 A, B; EV3 B, C, D, E; supplementary figures 1C, 2A, D-G.

The requested n info has been appended to the figure legends of 1D, 2B, 3B,C,D,E; EV2 A,B; EV3 B,C,D,E; SFig 1C, 2A,D-G.

6. Please note that the error bars are not defined in the legends of figures 3D, E; EV2 A, B; EV3 C, supplementary figures 2D-G.

All error bars now defined in the figure legends.

- Limit keywords to max. 5.

Reduced to 5

- Rename Extended Figure 1 etc. to Figure EV1 etc.

Renamed accordingly.

- Rename "Conflict of interests" to "Disclosure Statement & Competing Interests".

Renamed accordingly and statement has been updated.

- Indicate in legends exact n and exact p values, not a range, along with the statistical test used. To keep the figures "clear" some authors found providing an Appendix table Sx with all exact p-values preferable. You are welcome to do this if you want to.

Added to legends

- Please remove Reagents and Tools Table and uploaded it as a separate file.

Removed and now uploaded as a separate file

4) Appendix: Please remove appendix Patient history and fuse it with the one in the results as they are largely redundant. Please group the rest of appendix text in the section named "Appendix Supplementary Information" and update the callouts in the main manuscript text. Please rename appendix tables and figures to Appendix Table S1 and Appendix Figure S1 etc. and update their callouts in the main manuscript file.

Patient history consolidated within the main manuscript and redundancy removed from Appendix. Appendix Tables and Figures have been renamed. Appendix Supplementary Information callouts have been revised.

5) Funding: Please make sure that information about all sources of funding are complete in both

our submission system and in the manuscript. All sources of funding with the grant number listed in the Acknowledgement should be entered separately in our submission system.

Funding will be entered in the submission system.

6) The Paper Explained: Please provide "The Paper Explained" and add it to the main manuscript text. Please check "Author Guidelines" for more information.

<https://www.embopress.org/page/journal/17574684/authorguide#researcharticleguide>

The Paper Explained section has been added to the manuscript text.

7) Synopsis: Every published paper now includes a 'Synopsis' to further enhance discoverability. Synopses are displayed on the journal webpage and are freely accessible to all readers. They include separate synopsis image and synopsis text.

A Synopsis image has been provided as well as a Synopsis text document.

8) Source data: Please upload EV figure source data a single zip folder.

Done.

9) As part of the EMBO Publications transparent editorial process initiative (see our Editorial at <http://embomolmed.embopress.org/content/2/9/329>), EMBO Molecular Medicine will publish online a Review Process File (RPF) to accompany accepted manuscripts. This file will be published in conjunction with your paper and will include the anonymous referee reports, your point-by-point response and all pertinent correspondence relating to the manuscript. Let us know whether you agree with the publication of the RPF and as here, if you want to remove or not any figures from it prior to publication. Please note that the Authors checklist will be published at the end of the RPF.

We agree with publication of the RPF. In the previous "Response to Reviewers" document, we included two figures showing data of additional SETTLE patients that is not included in the manuscript. We request for all figures to be removed from the RPF.

10) Please provide a point-by-point letter INCLUDING my comments as well as the reviewer's reports and your detailed responses (as Word file).

Please accept this document as the Point by Point letter with responses.

I look forward to reading a new revised version of your manuscript as soon as possible.

Yours sincerely,

Zeljko Durdevic

Editor, EMBO Molecular Medicine

We thank you once again for considering our manuscript for publication in EMBO Molecular Medicine.

Yours sincerely,

Chinten James Lim, PhD

Philipp Lange, PhD

Chinten James Lim, PhD

Associate Professor, Department of Pediatrics, UBC Investigator, BC Children's Hospital Research Institute

Associate Director, Michael Cuccione Childhood Cancer Research Program

26th Feb 2025

Dear Dr. Lange,

We are pleased to inform you that your manuscript is accepted for publication and is now being sent to our publisher to be included in the next available issue of EMBO Molecular Medicine.

Zeljko Durdevic
Senior Editor
EMBO Molecular Medicine
